# FINE-GRAIN INFERENCE ON OUT-OF-DISTRIBUTION DATA WITH HIERARCHICAL CLASSIFICATION

## ABSTRACT

Machine learning methods must be trusted to make appropriate decisions in real-world environments, even when faced with out-of-distribution (OOD) samples. Many current approaches simply aim to detect OOD examples and alert the user when an unrecognized input is given. However, when the OOD sample significantly overlaps with the training data, a binary anomaly detection is not interpretable or explainable, and provides little information to the user. We propose a new model for OOD detection that makes predictions at varying levels of granularity—as the inputs become more ambiguous, the model predictions become coarser and more conservative. Consider an animal classifier that encounters an unknown bird species and a car. Both cases are OOD, but the user gains more information if the classifier recognizes that its uncertainty over the particular species is too large and predicts "bird" instead of detecting it as OOD. Furthermore, we diagnose the classifier's performance at each level of the hierarchy improving the explainability and interpretability of the model's predictions. We demonstrate the effectiveness of hierarchical classifiers for both fine- and coarse-grained OOD tasks.

## 1 INTRODUCTION

Real-world computer vision systems will encounter out-of-distribution (OOD) samples while making or informing consequential decisions. Therefore, it is crucial to design machine learning methods that make reasonable predictions for anomalous inputs that are outside the scope of the training distribution. Recently, research has focused on detecting inputs during inference that are OOD for the training distribution (Ahmed & Courville, 2020; Hendrycks & Gimpel, 2017; Hendrycks et al., 2019; Hsu et al., 2020; Huang & Li, 2021; Lakshminarayanan et al., 2017; Lee et al., 2018; Liang et al., 2018; Liu et al., 2020; Neal et al., 2018; Roady et al., 2020; Inkawhich et al., 2022). These methods typically use a threshold on the model's "confidence" to produce a binary decision indicating if the sample is in-distribution (ID) or OOD. However, binary decisions based on model heuristics offer little interpretability or explainability.

The fundamental problem is that there are many ways for a sample to be out-of-distribution. Ideally, a model should provide more nuanced information about how a sample differs from the training data. For example, if a bird classifier is presented with a novel bird species, we would like it to recognize that the sample is a bird rather than simply reporting OOD. On the contrary, if the bird classifier is shown an MNIST digit then it should indicate that the digit is outside its domain of expertise.

Recent studies have shown that fine-grained OOD samples are significantly more difficult to detect, especially when there is a large number of training classes (Ahmed & Courville, 2020; Huang & Li, 2021; Roady et al., 2020; Zhang et al., 2021; Inkawhich et al., 2021). We argue that the difficulty stems from trying to address two opposing objectives: learning semantically meaningful features to discriminate between ID classes while also maintaining tight decision boundaries to avoid misclassification on fine-grain OOD samples (Ahmed & Courville, 2020; Huang & Li, 2021). We hypothesize that additional information about the relationships between classes could help determine those decision boundaries and simultaneously offer more interpretable predictions.

To address these challenges, we propose a new method based on hierarchical classification. The approach is illustrated in fig. 1. Rather than directly outputting a distribution over all possible classes, as in a flat network, hierarchical classification methods leverage the relationships between classes to produce conditional probabilities for each node in the tree. This can simplify the classification

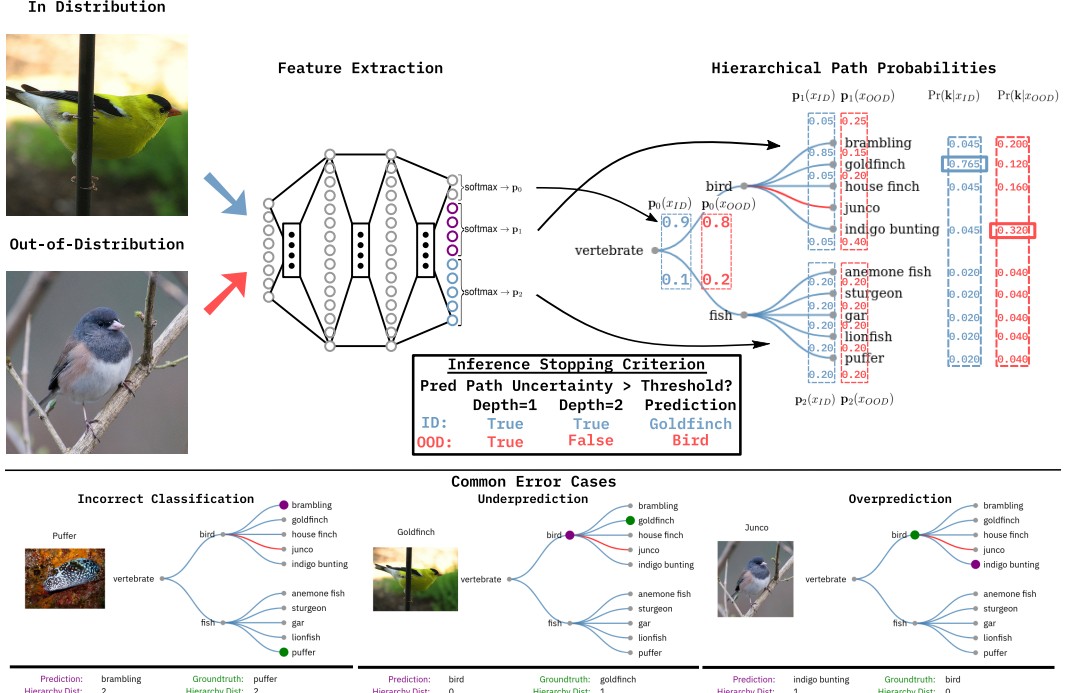

Figure 1: Method overview. *Top*: A ResNet50 extracts features from images and fully-connected layers output softmax probabilities $p_n(x_i)$ for each set in the hierarchy $\mathcal{H}$. Path-wise probabilities are used for final classification. Path-wise probability and entropy thresholds generated from the training set $\mathcal{D}_{\text{train}}$ form stopping criterion for the inference process. *Bottom*: Common error cases encountered by the hierarchical predictor. From left to right: Standard error results from and incorrect intermediate or leaf decision, ID under-prediction where the network predicts at a coarse granularity due to high uncertainty, OOD over-prediction where the OOD sample is mistaken for a sibling node.

problem since each node only needs to distinguish between its children, which are far fewer in number (Redmon & Farhadi, 2017; Ridnik et al., 2021). It can also improve the interpretability of the neural network (Wan et al., 2021). For example, we leverage these conditional probabilities to define novel OOD metrics for hierarchical classifiers and make coarser predictions when the model is more uncertain.

By employing an inference mechanism that predicts at different levels of granularity, we can estimate how similar the OOD samples are from the ID set and at what node of the tree the sample becomes OOD. When outliers are encountered, predicting at lower granularity allows the system to convey imprecise, but accurate information.

We also propose a novel loss function for the hierarchical softmax classification technique to address the fine-grained OOD scenario. We propose hierarchical OOD metrics for detection and create a hierarchical inference procedure to perform inference on ID and OOD samples, which improves the utility of the model. We evaluate the method's sensitivity to granularity under coarse- and fine-grain outlier datasets using ImageNet-1K ID-OOD holdout class splits (Russakovsky et al., 2015). Our in-depth analysis of the ID and OOD classification error cases illuminates the behavior of the hierarchical classifier and showcases how explainable models are instrumental for solving fine-grain OOD tasks. Ultimately, we show that hierarchical classifiers effectively perform inference on ID and OOD samples at varying granularity levels and improve interpretability in fine-grained OOD scenarios.

## 2 RELATED WORK

There are four main types of related works: those which focus on fine-grain out-of-distribution detection; those which emphasize scalability to large numbers of classes; those which leverage hierarchical classifiers; and those which utilize improved feature extactors.

**Fine-grain OOD.** Several recent works have identified poor OOD detection performance on fine-grain datasets with current methods (Ahmed & Courville, 2020; Roady et al., 2020; Zhang et al., 2021). Both (Zhang et al., 2021) and (Ahmed & Courville, 2020) highlight that selecting holdout classes from the original dataset is a more realistic OOD scenario that enforces detectors utilize semantically meaningful features to detect OOD samples rather than utilizing non-semantic differences between ID and OOD datasets to achieve good performance. Roady et al. (2020) also explore the issue of OOD granularity by evaluating performance across coarse- and fine-grained samples and outline a need for new methods to address fine-grained OOD detection. These previous two works emphasize the importance of OOD granularity especially when developing solutions for real-world applications. We expand upon these works by designing OOD detection datasets at varying levels of granularity. We provide an in-depth analysis of performance across OOD granularity.

**Scalable OOD.** Current OOD detection techniques struggle to scale to higher resolution images or to tasks with larger numbers of ID classes (Ahmed & Courville, 2020; Huang & Li, 2021; Roady et al., 2020). Huang & Li (2021) improve upon current OOD detection techniques' scalability by partitioning the classification task into smaller subsets of semantically similar categories, thereby reducing each decision's complexity. Grouping labels into exclusive subsets allows for an explicit "other" class to be added that is optimized to predict when the input is OOD for the current classifier. This approach uses the training data from all other subsets in a similar manner to how outlier exposure uses auxiliary outlier data (Hendrycks et al., 2019). However, Hendrycks et al. (2019) utilize an entropy based optimization for the outliers instead of an explicit "other" class. Huang & Li (2021) improve the scalability, but their approach is incapable of inference on ID and OOD data. Furthermore, Huang & Li (2021) choose 8 class groupings from WordNet without providing a systematic process of pruning the hierarchy. As the set of ID classes grows the relationships betweeen classes becomes more complex and the decision of how to group labels becomes increasingly arbitrary. We utilize the full WordNet hierarchy which provides natural boundaries between label subsets and systematically prune the hierarchy. Note that our method does not use auxiliary outlier data and developing a technique for improving uncertainty estimates via auxiliary data is left for future studies.

**Hierarchical classifiers.** Hierarchical classification methods have recently been exploited to improve the classifier's scalability to extremely large label space tasks (Ridnik et al., 2021) and to enhance deep learning explainability (Wan et al., 2021). The current work seeks to utilize these properties to improve OOD detection methods. Ridnik et al. (2021) utilize the WordNet hierarchy (Fellbaum, 1998) to train a classifier on Imagenet-21K including 11,221 ID classes aiming to improve pretrained model weight available for transfer learning. Wan et al. (2021) instead generate a binary decision tree from the learned weights of a standard softmax classifier using agglomerative clustering. Furthermore, Wan et al. (2021) thoroughly analyze the hierarchical classifier's behavior at each node in the tree to explain the network's prediction behavior. This method ensures that the hierarchical relationships are based on visual similarity instead of human defined semantic similarity. However, for OOD detection tasks we find binary trees generated from pretrained model weights do not perform as well as shallower hierarchies for OOD tasks (table 4). Furthermore, we build upon (Wan et al., 2021)'s interpretability study by utilizing the hierarchical structure to explain the common failure modes on ID and OOD data.

Another group of techniques incorporate hierarchical structures into the feature extractor (Ahmed et al., 2016; Yan et al., 2015) or training procedure (Alsallakh et al., 2018). In this work we do not consider specialized network architectures and training procedures because we are interested in comparing the performance of hierarchical vs. flat classification. We demonstrate in table 4 and in section 3.5 the advantages of hierarchical classification for interpretable OOD. Exploring custom hierarchical architectures is a promising area of future research to improve interpretable OOD methods.

**Improved feature extractors.** Vision transformers (Kolesnikov et al., 2021) and contrastive learning (Chen et al., 2020) methods learn improved feature extractors leading to gains in overall classification accuracy as well as transfer learning outcomes. Recently, (Tack et al., 2020; Winkens et al., 2020) utilize a SimCLR-based (Chen et al., 2020) architecture and (Ren et al., 2021; Vaze et al., 2022) utilize ViT (Kolesnikov et al., 2021) to improve feature representations and OOD performance. Likewise, our hierarchical method will likely mutually benefit from learning improved feature extractors. However, due to the significant computational resources required to train such models we leave integrating hierarchical classifiers with contrastive learning and vision transformers for future studies.

## 3 METHOD

We train a hierarchical softmax classifier (section 3.1) to generate path probabilities at all nodes in the hierarchy with a multi-objective loss function to optimize for classification performance and OOD detection (section 3.2). Path-wise probability and entropy scoring metrics replace softmax probabilities for OOD detection to incorporate model certainty along the prediction path (section 3.3). We employ these path-wise metrics as an inference stopping criterion to generate predictions on intermediate nodes (section 3.4). Finally, we measure hierarchical distance and accuracy to perform an in-depth analysis of model performance (section 3.5).

### 3.1 HIERARCHICAL CLASSIFICATION

We define a hierarchy, $\mathcal{H}$, as a tree-structured directed acyclic graph so that there is a unique path from the root node to each leaf node. For notation, associate each node in the tree with an integer $\{0, 1, \ldots, N\}$ where $0$ denotes the root node. Let $\mathsf{par}(n) \in \{0, \ldots, n-1\}$ denote the parent of node $n$, let $\mathsf{anc}(n) \subset \{0, \ldots, n-1\}$ be the set of all ancestors of node $n$, and let $\mathsf{ch}(n) \subseteq \{n+1, \ldots, N\}$ denote the set of children of node $n$. Finally, let $\mathcal{Y} \subset \{0, 1, \ldots, N\}$ denote the set of leaf nodes (i.e. nodes for which $\mathsf{ch}(n) = \emptyset$) and $\mathcal{Z} = \{0, 1, \ldots, N\} \setminus \mathcal{Y}$ be the set of internal nodes.

Each training data point has an input $x_i \in \mathbb{R}^d$ and a label $y_i \in \mathcal{Y}$, which is associated with a leaf node of the hierarchy. The training distribution, $\mathcal{D}_{\mathsf{train}} = \{(x_i, y_i)\}$, is comprised of tuples of input images, $x_i$, and associated leaf nodes, $y_i$. For each node $n$ in the set of internal nodes $\mathcal{Z}$, we define $\mathcal{D}_n \subseteq \mathcal{D}_{\mathsf{train}}$ to be all the samples $(x_i, y_i)$ whose ancestors contain $n$. Likewise, define $\mathcal{D}_{\neg n}$ all the examples that whose ancestors do not contain $n$.

Given the input $x_i$, the network outputs probability distributions $\mathbf{p}_n = [p_{n,1}, \ldots, p_{n,|\mathsf{ch}(n)|}]$ for each internal node $n$, where $p_{n,j} \geq 0$ and $\sum_{j=1}^{|\mathsf{ch}(n)|} p_{n,j} = 1$. In practice, we model each $\mathbf{p}_n$ as a softmax function of the features in the penultimate layer of a neural network. We parameterize a distribution on leaf nodes as the product of probabilities associated with each node along that path,

$$\Pr(y_i = k \mid x_i) = \prod_{a \in \mathsf{anc}(k) \setminus 0} p_{\mathsf{par}(a), a}. \tag{1}$$

As noted by (Wan et al., 2021), path-wise probabilities allow for errors at intermediate nodes to be corrected as opposed to making "hard" decisions at each level of the tree and following the path to the prediction. Note that the path probabilities in eq. (1) form a proper probability distribution such that $\sum_{k \in \mathcal{Y}} \Pr(y_i = k \mid x_i) = 1$. At test time, we take the network's prediction to be the leaf node with the highest probability, $\hat{y}_i = \arg\max_{k \in \mathcal{Y}} \Pr(y_i = k \mid x_i)$.

### 3.2 HIERARCHICAL OOD LOSS

To achieve high ID accuracy and reliable OOD detection we propose a weighted multi-objective loss to optimize the hierarchical classifier. Formally, it is defined as,

$$\mathcal{L}_{\mathsf{soft}} = \sum_{n \in \mathcal{Z}} W_n \cdot \sum_{(x,y) \in \mathcal{D}_n} H\left[\mathsf{onehot}_n(y), \mathbf{p}_n(x)\right] \tag{2}$$

$$W_n = \frac{|\{j \in \{1 \ldots N\} : n \in \mathsf{anc}(j)\}|}{N} \tag{3}$$

$$\mathcal{L}_{\mathsf{other}} = \sum_{n \in \mathcal{Z}} \sum_{(x,y) \in \mathcal{D}_{\neg n}} H\left[\mathcal{U}(|\mathsf{ch}(n)|), \mathbf{p}_n(x)\right] \tag{4}$$

$$\mathcal{L} = \alpha \cdot \mathcal{L}_{\mathsf{soft}} + \beta \cdot \mathcal{L}_{\mathsf{other}}, \tag{5}$$

where $H[p, q]$ is the cross-entropy from $p$ to $q$ and $\mathsf{par}_n(y)$ is the one-hot vector for the ancestor of $y$ corresponding to node $n$, $\mathsf{onehot}_n(y) = [\mathbb{1}(k \in \mathsf{anc}(y)) : k \in \mathsf{ch}(n)]$.

The first objective optimizes the network for ID classification accuracy by applying cross-entropy to the network's predictions $\Pr(\mathbf{k}|x_i) = [\Pr(y_i = k|x_i)]_{k \in \mathcal{Y}}$ for each sample in the training distribution,

$\mathcal{D}_{\text{train}}$ (eq. (2)) [1]. The second objective (eq. (4)) drives the probabilities at internal nodes that are not along the path from root to ground-truth node to the uniform distribution, parameterized by size, ($\mathcal{U}(s)$) with cross-entropy. This utilizes in-distribution data as outliers for all nodes in the hierarchy that are not one of its ancestors.

As in Ridnik et al. (2021), we find that weighting the cross-entropy contribution of each internal node improves optimization. We weight each node's loss, eq. (3), based on the proportion of the total number of classes ($N$) which are children of the node. This assigns higher weight to nodes that are closer to the root of the tree as these decisions have the greatest effect downstream classification performance.

### 3.3 PREDICTION PATH ENTROPY OOD METRIC

We propose prediction path based OOD scoring functions for performing OOD detection with hierarchical classifiers. First, we propose using maximum prediction path probabilities calculated according to eq. (1) which is hierarchical analog to max softmax probability for standard networks. Second, we propose using either the mean path-wise entropy, $H_{\text{mean}}$, or the maximum path-wise entropy, $H_{\text{max}}$, as metrics for OOD detection. Formally, these are defined as,

$$H_{\text{mean}}(x_i) = \frac{1}{|\mathsf{anc}(\hat{y}_i)|} \sum_{n \in \mathsf{anc}(\hat{y}_i)} H[\mathbf{p}_n] \tag{6}$$

$$H_{\text{max}}(x_i) = \max_{n \in \mathsf{anc}(\hat{y}_i)} H[\mathbf{p}_n], \tag{7}$$

### 3.4 INFERENCE STOPPING CRITERION

Given a hierarchical classifier optimized over $\mathcal{D}_{\text{train}}$, we define a stopping criterion utilizing the performance statistics on the validation data. Specifically, we select a true negative rate (TNR) on the ID data to decide our inference stopping threshold from the micro-averaged receiver operating characteristic (ROC) curve. In practice, this TNR threshold will be determined by the specific application's prediction fidelity requirements. Micro-averaged ROC curves are used to generate the TNR thresholds for each node in $\mathcal{Z}$. We utilize path probabilities $\Pr(n|x_i)$ as the threshold score.

During inference the leaf node prediction $\hat{y}$ is determined, then the prediction path $\mathsf{anc}(\hat{y})$ is traversed from root to leaf. If any of the nodes in the path do not meet the TNR threshold, the parent node is chosen as the prediction (fig. 1). Both global path probability and node-wise probability and mean-, min-entropy were explored as TNR threshold metrics.

### 3.5 HIERARCHICAL ACCURACY AND DISTANCE

We analyze the hierarchical classifier's inference on ID and OOD samples with top-1 accuracy, as well as, average hierarchical distance. The groundtruth for OOD samples is the closest ancestor that is contained within ID hierarchy. For example, the OOD node *junco* in 1 is assigned the ID groundtruth node *bird*.

Furthermore, we consider the inference procedure's failure modes by decomposing the hierarchy distance into two parts: (1) the prediction and (2) the groundtruth distance to their closest common parent. Hierarchy distance is defined as the number of edges in the hierarchy between two nodes. By recording the groundtruth and prediction distances to the closest common parent we can determine how frequently the model incorrectly predicts, overpredicts, and underpredicts for a set of inputs. fig. 1 (bottom) depicts common error cases that are encountered and their corresponding hierarchy distances.

## 4 EXPERIMENTS

### 4.1 OOD DATASETS

---

[1] When $W_n = 1 \forall n \in \mathcal{Z}$ the form in eq. (2) is equivalent to the entropy over the leaf nodes $H[y, \Pr(k|\mathcal{D}_{\text{train}})]$

**Fine-grain OOD datasets.** Some applications may face more extreme OOD examples than others. To construct OOD detection tasks with varying degrees of difficulty, we leveraged the fact that the Imagenet-1K classes correspond to nouns in the WordNet hierarchy (Fellbaum, 1998). We generated OOD sets by holding out subsets of Imagenet-1K classes in entire subtrees of the WordNet hierarchy. Withholding large subtrees—those rooted at low depths of the hierarchy—leads to coarse-grained OOD detection tasks, since the held-out classes are very different from the training classes. Holding out small subtrees—those rooted at nodes deep in the hierarchy—leads to fine-grained OOD-detection tasks. We created 2 datasets Imagenet-100 and Imagenet-1K starting from a 100 class subset of Imagenet-1K classes and the full Imagenet-1K dataset. Table 2 provides summary statistics of both Imagenet datasets.

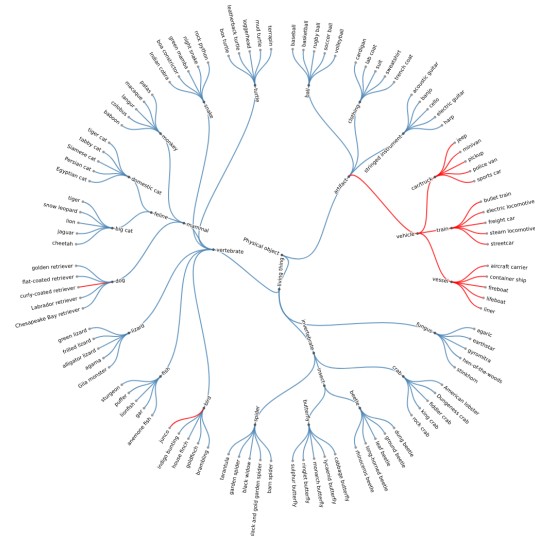

Figure 2: IMAGENET-100 pruned WordNet hierarchy. Red edges correspond to OOD paths and blue to ID.

We used the 100 class dataset, Imagenet-100, to evaluate the effects of additional fully-connected (FC) layers and hierarchy choice. We prune the WordNet hierarchy for this dataset by removing all nodes with a single child and manually combining semantically similar leaf nodes into groups of 5. Figure 2 shows the hierarchy with randomly chosen coarse- and fine-grain OOD classes in red and the training classes in blue.

We used Imagenet-1K to evaluate the hierarchical classifier's scalability to large ID datasets. Again, we prune the WordNet hierarchy by first removing all nodes with a single child. We then further pruned the hierarchy to a fixed number of internal nodes by iteratively pruning the node with the minimum entropy by merging its children. We use training distribution statistics to calculate the entropy for each internal node. However, we find that these simplified hierarchies do not improve ID accuracy or OOD performance.

**Coarse-grain OOD datasets.** For further comparison to other baseline methods and analysis of far-OOD data we used coarse-grain OOD datasets for baseline comparisons. Specifically, we compared to the iNaturalist (Horn et al., 2018), SUN (Xiao et al., 2010), Places365 (Zhou et al., 2018), and Textures (Cimpoi et al., 2014) subsets from (Wan et al., 2021).

## 4.2 MODEL TRAINING

All models were trained from scratch as the available pretrained weights were trained on the fine-grained OOD holdout classes. We used a ResNet50 (He et al., 2016) backbone for all models and trained for 90 epochs of stochastic gradient descent (SGD). We used a learning rate of 0.1 with learning rate decay steps with a decay factor of 0.1 performed at epoch 30 and 60. The momentum and weight decay parameters were 0.9 and $10^{-4}$, respectively. We standardized the training hyperparameters to avoid performance differences due to the optimization procedure.

## 4.3 RESULTS

**Fine-grain OOD performance** We found that the hierarchical softmax classifier (HSC) outperformed baseline methods on the Imagenet-100 dataset and performs within 2% of the best performing baseline for Imagenet-1K dataset (table 6). The hierarchical classifiers benefit from additional FC layers on top of the ResNet50 feature extractor due to the additional complexity of classifying into several disjoint sets (table 7). This also suggests that specialized hierarchical network architectures (Ahmed et al., 2016; Yan et al., 2015; Alsallakh et al., 2018) that learn features specific to each node's classifier may further improve OOD performance. We assessed the effect of holdout class granularity and found that the softmax-based OOD heuristics (MSP, ODIN, and prediction path probability) are most sensitive to fine-grain OOD samples whereas MOS and path entropy metrics perform best on coarse-grain OOD as shown in table 6. Also, we find that outlier exposure improves coarse-grain

Table 1: Hierarchical softmax classifier (HSC) performance on the Imagenet-100 and Imagenet-1K datasets. The $\mathcal{L}_{\text{soft}}$ and $\mathcal{L}_{\text{other}}$ weights ($\alpha$, $\beta$) and the OOD metric are given in parenthesis for each HSC model. OOD performance is measured by AUROC scores for the fine-, medium- and coarse-OOD classes as well as the overall OOD performance. The best performing baseline and HSC models are bolded. Each cell includes the performance statistics across 3 models trained with separate random seeds. For ensemble OOD methods (Lakshminarayanan et al., 2017) cells follow the format: "mean(std)/ensemble". Note that relative Mahalanobis Ren et al. (2021) performance is reported as it outperformed the original method (Lee et al., 2018). All models are ResNet50 architectures trained for 90 epochs. All numbers are percentages.

| MODEL (METHOD) | ACCURACY | AUROC | | | |
| --- | --- | --- | --- | --- | --- |
| | | FINE | MEDIUM | COARSE | OVERALL |
| IMAGENET 100 | | | | | |
| MSP (HENDRYCKS & GIMPEL, 2017) | 81.26(0.53)/82.75 | 72.47(0.31)/73.62 | — | 92.62(0.67)/94.38 | 90.25(0.62)/91.94 |
| ODIN (LIANG ET AL., 2018) | 81.26(0.53)/82.75 | 72.93(1.87)/74.36 | — | 95.90(0.47)/**96.71** | 93.20(0.36)/**94.08** |
| MAHALANOBIS (REN ET AL., 2021) | 81.26(0.53) | **78.05**(0.09) | — | 91.34(0.62) | 89.78(0.54) |
| MOS (HUANG & LI, 2021) | **82.41**(0.02) | 70.00(0.72) | — | **96.66**(0.23) | **93.66**(0.22) |
| HSC ($\alpha = 1$, $\beta = 0$, PRED) | 82.38(0.06)/83.25 | 76.78(3.38)/79.80 | — | 93.93(0.22)/95.08 | 91.33(0.28)/92.38 |
| HSC ($\alpha = 1$, $\beta = 0$, $H_{\text{mean}}$) | 82.38(0.06)/83.25 | 77.27(3.83)/75.86 | — | 96.90(0.11)/96.89 | 93.92(0.20)/93.17 |
| HSC ($\alpha = 1$, $\beta = 0.2$, PRED) | **82.85**(0.14)/**84.05** | 79.40(0.76)/80.67 | — | 95.06(0.13)/96.05 | 92.29(0.15)/93.33 |
| HSC ($\alpha = 1$, $\beta = 0.2$, $H_{\text{mean}}$) | **82.85**(0.14)/**84.05** | **79.40**(0.67)/76.35 | — | **97.23**(0.11)/96.93 | **94.08**(0.13)/93.30 |
| IMAGENET 1K | | | | | |
| MSP (HENDRYCKS & GIMPEL, 2017) | 74.94(0.08)/77.05 | 74.30(0.24)/74.90 | 79.33(0.17)/81.32 | 80.42(0.19)/82.71 | 77.96(0.11)/79.57 |
| ODIN (LIANG ET AL., 2018) | 74.94(0.08)/**77.05** | **76.25**(0.11)/**77.62** | **79.84**(0.21)/**81.82** | 81.95(0.15)/84.02 | **79.18**(0.13)/**80.98** |
| MOS (HUANG & LI, 2021) | **75.00**(0.43) | 74.71(0.90) | 74.00(0.53) | **87.11**(0.40) | 77.32(0.59) |
| HSC ($\alpha = 1$, $\beta = 0$, PRED) | 73.79(0.13)/76.51 | 72.73(0.47)/73.42 | 78.33(0.27)/80.49 | 80.64(0.17)/82.92 | 77.07(0.24)/78.78 |
| HSC ($\alpha = 1$, $\beta = 0$, $H_{\text{mean}}$) | 73.79(0.13)/76.51 | 64.84(0.64)/61.45 | 77.03(0.28)/75.02 | 82.86(0.11)/85.43 | 74.47(0.31)/73.09 |
| HSC ($\alpha = 1$, $\beta = 0.05$, PRED) | **74.46**(0.06)/**76.79** | **72.86**(0.56)/**73.69** | **79.40**(0.29)/**81.45** | 82.38(0.58)/84.35 | **77.99**(0.41)/**79.63** |
| HSC ($\alpha = 1$, $\beta = 0.05$, $H_{\text{mean}}$) | **74.46**(0.06)/**76.79** | 63.54(0.74)/61.56 | 76.62(0.35)/75.44 | **84.54**(0.40)/**86.01** | 74.27(0.34)/73.45 |

OOD performance across all HSC metrics. Finally, we find that the Imagenet-100 trained ODIN detector is the best performer on the 4 far-OOD datasets table 9. However, MOS is the best performer on iNaturalist, SUN and Places when scaling the number of ID classes to the Imagenet-1K dataset.

**Hierarchy selection.** In table 4, we evaluate the sensitivity to hierarchy depth and composition for Imagenet-100 datasets with two human-defined semantic, WordNet (Fellbaum, 1998) based hierarchies and a visually-derived binary decision tree induced from the learned weights of a standard softmax classifier following the procedure from (Wan et al., 2021). The two-level hierarchy is created using the direct parents of the leaf-nodes from WordNet. We find that the performance across all OOD metrics introduced in section 3.3 is comparable. In particular, there is no apparent benefit to visually-derived hierarchies vs. human-defined semantic hierarchies. However, we believe that the hierarchy is a critical design choice and is likely application dependent. Specifically, the hierarchy's class balance, depth, and alignment with visual features are important characteristics to consider. In natural image classification domains, human-defined semantic structures may improve interpretability because they project image inputs into a human conceptual framework even though they may not perfectly represent the visual properties of the input.

**Scalability to large ID datasets.** In table 6, we find that HSC classifiers are able to scale to large ID datasets. However, the performance of the path-wise entropy based OOD detection metric $H_{\text{mean}}$ under-performs. We suspect that this can be attributed to deep, imbalanced hierarchies where some branches receive extremely low weights making optimization more difficult (eq. (3)). Indeed, we find that simplifying the hierarchy with custom pruning and a 2 level hierarchy we can regain the lost OOD performance and even outperform the best baseline table 5. Therefore, well balanced hierarchies and models with sufficient capacity avoid this optimization problem.

## 5 ANALYSIS

Hierarchical classifiers decompose the classification problem into simpler intermediate tasks. By analyzing the model's confidence at each intermediate decision, we can understand where the model becomes uncertain. Wan et al. (2021) show that through analyzing intermediate decisions we can explain the model's decision process to understand where the model makes mistakes and how it behaves on ambiguous labels, and we can use that insight to improve human trust in the predic-

tor. Therefore, hierarchical classifiers may greatly improve the interpretability and explainability compared to softmax classifiers.

We build off of this work by leveraging intermediate model confidence estimates to determine at what level of granularity to make a prediction. For ID data this corresponds to making more conservative predictions when the model is uncertain, based on the intuition that a correct, less specific prediction improves the user's confidence in a model than an incorrect, more specific prediction. Similarly, for OOD data the same method can indicate exactly where the sample diverges from the training distribution and therefore can predict the sample's parent class, allowing the user to make more informed decisions in the face of novel inputs.

### 5.1 THRESHOLDING TO REDUCE FALSE-POSITIVES

First, we aim to understand the effects of OOD data on the hierarchical classifier's performance and if thresholding is effective for detecting and predicting OOD samples. We plot the micro-ROC curves (fig. 3) for 4 synsets each corresponding to a separate classification decision in the hierarchy. The "artifact", "dog" and "bird" synsets include one or more OOD samples and the "ball" synset does not have any corresponding OOD samples (see fig. 2). Notice in Figure 3 that when adding the activations of the OOD data ("OOD" curve) the number of false-positives increases and AUROC drops compared to the ID-only curve because the OOD data is being predicted more confidently than some ID data. This occurs across all synsets in the hierarchy even in the "ball" synset that does not contain any OOD descendants. However, when we employ a path-wise probability based threshold at 99% TNR on the training data ("THR" curves in fig. 3), the performance is recovered in all synsets. The micro-ROC curves for all synsets is displayed in fig. 13.

### 5.2 OOD INFERENCE PERFORMANCE

Next, we compare path-wise and node-wise thresholding methods (section 3.4). In path-wise thresholding, a global uncertainty threshold is set and the prediction is made at the deepest node that meets the threshold. In the node-wise method, a threshold is set for each internal node to determine if the node is certain enough to make a prediction.

On the Imagenet-100 dataset, we achieve 73% accuracy on the OOD samples while maintaining 74% ID accuracy using a path-wise probability threshold chosen at 95% TNR as witnessed by the blue line in Figure 10. In fig. 4, we plot the ID and OOD hierarchy distance over the TNRs used to set the prediction threshold to analyze the inference method's behavior. Note that the ODIN baseline is only capable of predicting a leaf-node or not predicting at all. This leads to the larger OOD hierarchy

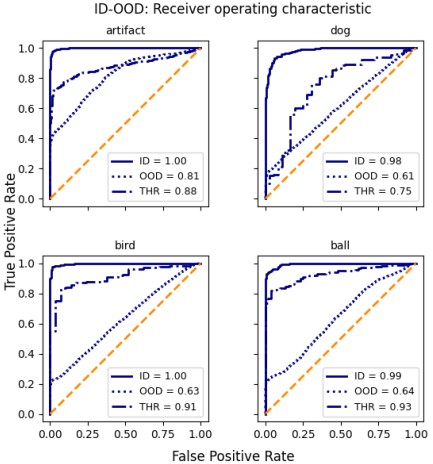

Figure 3: Imagenet-100 synset micro-ROC curves for ID data only, ID and OOD, and ID and OOD with a TNR=0.95 path threshold.

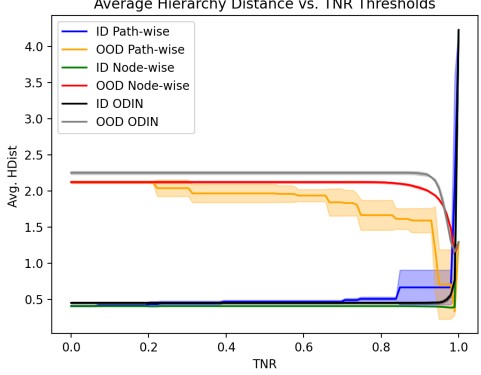

Figure 4: Imagenet-100 ID and OOD average hiearchy distance across TNR threshold values for path-wise and node-wise threshold metrics with ODIN baseline.

distance across the TNR spectrum (fig. 4 grey), as well as, the faster decline in ID performance compared to the node-wise thresholding method (fig. 4 black vs green). Figure 10 further shows the limitations of binary OOD detectors as they are incapable of predicting on OOD samples.

The large step changes and deviation of path-wise thresholding in Figure 10 and Figure 4 reflect that the path-wise thresholds cause the network to predict at increasingly coarse nodes as the confidence degrades with increasing depth (i.e. specificity, see eq. (1)). Whereas the behavior of the node-wise thresholding technique changes the required certainty at all nodes in the hierarchy leading to smoother behavior and tighter standard deviations in the green and red lines in Figure 4 and Figure 10. When the distribution of OOD classes is balanced across granularity levels, the node-wise inference technique greatly outperforms the path-wise technique due to the step wise nature of the path-wise technique (figs. 6 to 9).

Accuracy is a binary indicator of performance for each sample which is well suited to softmax and other flat classifiers that do not capture the relationships between classes. However, for inference on OOD data and prediction under uncertainty, hierarchy distance is useful as a measure of the semantic differences between the prediction is to the groundtruth. We show that OOD average hierarchy distance consistently decreases and the ID average hierarchy distance remains relatively constant (fig. 4, bottom). While the ID accuracy drops from 82.75% to 74.46% at the 95% TNR node-wise threshold, the average hierarchy distance decreases from 0.4045 to 0.4005 (fig. 4 bottom right). This indicates that the increase in hierarchy distance from the ∼8% drop in ID accuracy is counteracted by predicting the ∼17% of original error cases at a lower specificity, thereby reducing the average hierarchy distance. Therefore, by allowing the hierarchical classifier to predict with less specificity, we can improve the overall prediction quality by removing uncertain leaf node predictions.

## 5.3 Inference Error Analysis

Error analysis is crucial to interpreting deep learning methods (Wan et al., 2021). We find that analyzing the hierarchy distance between prediction and groundtruth nodes for ID and OOD samples illuminates the common pathologies of our inference method fig. 12. Specifically, we find that both the path- and node-wise OOD errors commonly occur from over-prediction indicated by the concentration of samples along the prediction axis with a ground-truth distance of 0. Conversely, the ID data is generally under-predicted by 1 node indicating that the most uncertain ID samples are being predicted at a coarser granularity fig. 12. In the standard ROC based metrics, the OOD samples with a greater certainty than the leaf-level ID predictions would be considered false-positives. We believe that analyzing the hierarchical distance is a more appropriate compared to standard AUROC metrics for fine-grained OOD scenarios as it allows for predicting uncertain ID samples at a coarser level which is likely the desired behavior when the ID and OOD are very closely related.

## 6 Conclusion

In this work, we propose a method for performing inference on OOD samples with hierarchical classifiers. We argue that in the fine-grain OOD scenario, detecting a sample as OOD is often not the optimal behavior when the sample is a descendant of an ID class. Hierarchical inference also improves ID prediction when the model's uncertainty is large and a more certain, less precise prediction is available. Hierarchical classification enforces that intermediate predictions are made that can be directly analyzed to better interpret, explain, and validate the model's decisions prior to deployment. We provide empirical evidence to support these claims and consider hierarchical inference to be a promising direction for future research for creating trusted real-world machine learning systems.

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

# A APPENDIX

## A.1 COMPUTE RESOURCES

All experiments were run on an internal compute cluster with nodes containing 8 NVIDIA RTX A5000 GPUs. Imagenet 100 experiments were trained on 1 GPU for 90 epochs which completed in ∼8 hours for the longest running experiments. Imagenet 1K experiments were trained on 2 GPUs with data parallelization. Training for the Imagenet 1K's longest running experiments lasted ∼52.5 hours.

## A.2 HIERARCHY STATISTICS

Table 2: Imagenet dataset holdout set statistics. The number of leaf nodes that are held out due to trimmed branches at each level of granularity. The uniform probability used to choose the holdout nodes and the hierarchy depths for each granularity level are given for each dataset.

| DATASET | MAX DEPTH INTERNAL LEAFS | # LEAF HOLDOUTS | | |
|---|---|---|---|---|
| | | COARSE | MEDIUM | FINE |
| IMAGENET 100 | 6 | 15 | 0 | 2 |
| | 28 | — | — | — |
| | 100 | LVLS 2 | — | LEAFS |
| BALANCED IMAGENET 100 | 6 | 15 | 5 | 10 |
| | 28 | — | — | — |
| | 100 | LVLS 2 | 4–5 | LEAFS |
| IMAGENET 1K | 15 | 74 | 101 | 54 |
| | 369 | P=0.25 | 0.0625 | 0.0125 |
| | 1000 | LVLS 3–6 | 7–10 | 11–15 |

## A.3 BALANCED IMAGENET 100 RESULTS

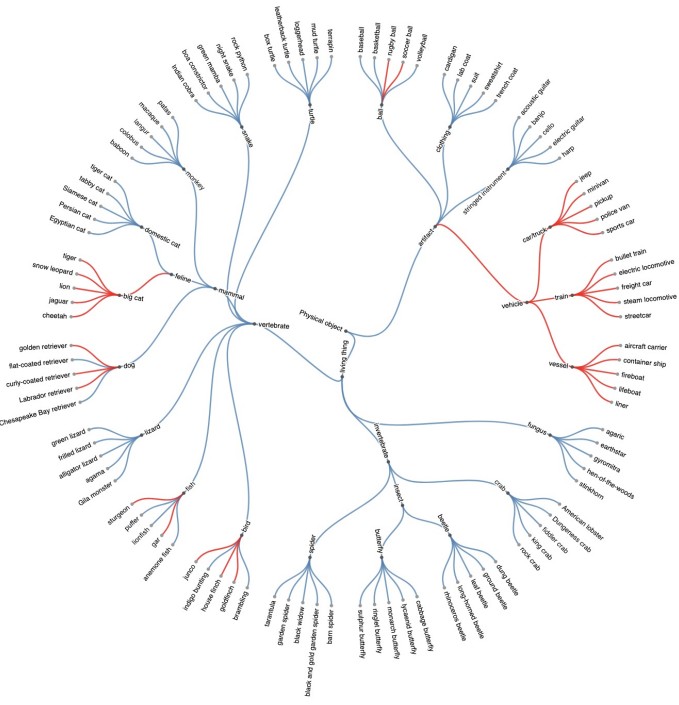

Figure 5: BALANCED IMAGENET 100 pruned WordNet hierarchy. Red edges correspond to OOD paths and blue to ID.

Table 3: Hierarchical softmax classifier (HSC) performance on the Balanced Imagenet 100 dataset. The $\mathcal{L}_{\text{soft}}$ and $\mathcal{L}_{\text{other}}$ weights ($\alpha$, $\beta$) and the OOD metric are given in parenthesis for each HSC model. OOD performance is measured by AUROC scores for the fine-, medium- and coarse- OOD classes as well as the overall OOD performance. Each cell includes the performance statistics across 3 models trained with separate random seeds. For ensemble OOD methods (Lakshminarayanan et al., 2017) cells follow the format: "mean(std)/ensemble". Note that relative Mahalanobis Ren et al. (2021) performance is reported as it outperformed the original method (Lee et al., 2018). All models are ResNet50 architectures trained for 90 epochs. All numbers are percentages.

| Model (Method) | Accuracy | AUROC | | | |
| --- | --- | --- | --- | --- | --- |
| | | Fine | Medium | Coarse | Overall |
| Balanced Imagenet 100 | | | | | |
| MSP (Hendrycks & Gimpel, 2017) | **80.85**(0.23)/**82.11** | 72.11(0.65)/73.91 | 71.07(0.58)/73.51 | 92.32(0.49)/93.66 | 82.04(0.28)/83.72 |
| ODIN (Liang et al., 2018) | **80.85**(0.23)/**82.11** | 79.16(0.56)/80.37 | 74.35(0.57)/75.84 | **96.09**(0.63)/**96.78** | **86.82**(0.23)/**87.82** |
| Energy (Liu et al., 2020) | **80.85**(0.23)/**82.11** | 79.30(0.59)/80.48 | 74.22(0.57)/75.35 | 95.97(0.71)/96.77 | 86.79(0.26)/87.77 |
| Mahalanobis (Ren et al., 2021) | **80.85**(0.23) | **83.07**(0.80) | 72.66(0.59) | 91.11(0.89) | 85.36(0.64) |
| MOS (Huang & Li, 2021) | 80.35(0.21) | 81.49(0.65) | **86.80**(0.35) | 74.23(1.05) | 86.80(0.35) |
| HSC ($\alpha = 1, \beta = 0$, PRED) | 81.19(0.26)/81.83 | 69.44(0.90)/71.25 | 71.57(1.44)/73.11 | 93.29(0.18)/94.49 | 81.72(0.44)/83.18 |
| HSC ($\alpha = 1, \beta = 0$, $H_{\text{mean}}$) | 81.19(0.26)/81.83 | 69.81(1.01)/78.56 | 68.12(1.61)/72.75 | 96.46(0.11)/96.69 | 82.85(0.60)/86.66 |
| HSC ($\alpha = 1, \beta = 0$, $H_{\text{max}}$) | 81.19(0.26)/81.83 | 66.57(0.76)/78.86 | 71.00(1.53)/73.44 | 89.46(0.23)/94.39 | 78.75(0.60)/85.72 |
| HSC ($\alpha = 1, \beta = 0$, $H_{\text{min}}$) | 81.19(0.26)/81.83 | 70.72(1.85)/73.56 | 28.15(2.41)/71.45 | 95.21(0.79)/95.56 | 75.87(0.98)/84.21 |
| HSC ($\alpha = 1, \beta = 0$, ENERGY) | 81.19(0.26)/81.83 | 75.88(2.03)/70.26 | 70.02(3.12)/61.97 | **98.28**(0.11)/98.24 | 86.10(1.20)/82.87 |
| HSC ($\alpha = 1, \beta = 0.2$, PRED) | **81.83**(0.10)/**82.97** | 73.91(1.04)/75.52 | **73.88**(1.19)/**75.86** | 94.20(0.09)/95.32 | 84.05(0.46)/85.47 |
| HSC ($\alpha = 1, \beta = 0.2$, $H_{\text{mean}}$) | **81.83**(0.10)/**82.97** | 74.23(1.01)/80.38 | 70.64(1.33)/74.54 | 96.65(0.03)/96.43 | 84.84(0.42)/87.43 |
| HSC ($\alpha = 1, \beta = 0.2$, $H_{\text{max}}$) | **81.83**(0.10)/**82.97** | 71.29(0.89)/80.74 | 73.18(1.02)/74.23 | 92.68(0.14)/95.12 | 82.30(0.44)/86.84 |
| HSC ($\alpha = 1, \beta = 0.2$, $H_{\text{min}}$) | **81.83**(0.10)/**82.97** | 73.72(1.98)/**81.94** | 27.26(0.85)/75.26 | 95.76(0.37)/96.33 | 77.00(0.44)/**88.02** |
| HSC ($\alpha = 1, \beta = 0.2$, ENERGY) | **81.83**(0.10)/**82.97** | **80.03**(1.20)/73.79 | 73.50(2.09)/64.63 | 98.01(0.04)/**98.62** | **87.93**(0.63)/84.68 |

## A.4 Imagenet 100 Hierarchy Experiments

Table 4: ID and OOD sensitivity to hierarchy selection on the Imagenet-100 dataset. $\mathcal{H}$ type indicates whether the hierarchy is defined by human semantics or learned visual feature clustering. All numbers are percentages.

| Hierarchy $\mathcal{H}$ | $\mathcal{H}$ Type | Accuracy | Path Predition | Path Entropy | | |
| --- | --- | --- | --- | --- | --- | --- |
| | | | | Mean | Max | Min |
| 2 Lvl WN | Semantic | 82.19(0.38) | 91.73(0.17) | 93.43(0.08) | 92.12(0.04) | 93.08(0.13) |
| Pruned WN | Semantic | 82.38(0.06) | 91.33(0.28) | 93.92(0.20) | 89.16(0.46) | 93.70(0.13) |
| Binary NBDT (Wan et al., 2021) | Visual | 81.28(0.48) | 91.33(0.29) | 92.92(0.14) | 86.68(0.18) | 93.15(0.24) |

## A.5 Imagenet 1K Hierarchy Experiments

Table 5: Effect of hierarchy selection on Imagenet-1K dataset. $\mathcal{H}$ type indicates whether the hierarchy is defined by human semantics or learned visual feature clustering. Note all models are HSC with $\alpha$ and $\beta$ in parentheses. All numbers are percentages.

| Hierarchy $\mathcal{H}$ | $\mathcal{H}$ Type | Accuracy | Path Predition | Path Entropy | | |
| --- | --- | --- | --- | --- | --- | --- |
| | | | | Mean | Max | Min |
| Full ($\alpha = 1, \beta = 0.05$) | Semantic | 74.34(0.03) | 77.73(0.25) | 74.21(0.35) | 76.68(0.09) | 62.04(0.75) |
| Custom Prune ($\alpha = 1, \beta = 0.05$) | Semantic | 74.12(0.05) | 78.24(0.17) | 71.31(0.16) | 75.97(0.15) | 60.95(0.30) |
| 2 Lvl ($\alpha = 1, \beta = 0.1$) | Semantic | 75.03(0.10) | 78.94(0.24) | **79.76**(0.28) | 79.52(0.22) | 69.91(1.43) |

## A.6 IMAGENET 100 AND IMAGENET 1K ADDITIONAL OOD METRICS

Table 6: Hierarchical softmax classifier (HSC) performance on the Imagenet-100 and Imagenet-1K datasets with additional OOD metrics. The $\mathcal{L}_{\text{soft}}$ and $\mathcal{L}_{\text{other}}$ weights ($\alpha$, $\beta$) and the OOD metric are given in parenthesis for each HSC model. OOD performance is measured by AUROC scores for the fine-, medium- and coarse- OOD classes as well as the overall OOD performance. The best performing baseline and HSC models are bolded. Each cell includes the performance statistics across 3 models trained with separate random seeds. For ensemble OOD methods (Lakshminarayanan et al., 2017) cells follow the format: "mean(std)/ensemble". Note that relative Mahalanobis Ren et al. (2021) performance is reported as it outperformed the original method (Lee et al., 2018). All models are ResNet50 architectures trained for 90 epochs. All numbers are percentages.

| MODEL (METHOD) | ACCURACY | AUROC | | | |
| --- | --- | --- | --- | --- | --- |
| | | FINE | MEDIUM | COARSE | OVERALL |
| IMAGENET 100 | | | | | |
| MSP (HENDRYCKS & GIMPEL, 2017) | 81.26(0.53)/82.75 | 72.47(0.31)/73.62 | — | 92.62(0.67)/94.38 | 90.25(0.62)/91.94 |
| ODIN (LIANG ET AL., 2018) | 81.26(0.53)/82.75 | 72.93(1.87)/74.36 | — | 95.90(0.47)/**96.71** | 93.20(0.36)/**94.08** |
| ENERGY (LIU ET AL., 2020) | 81.26(0.53)/82.75 | 72.76(1.99)/74.00 | — | 95.72(0.50)/96.63 | 93.02(0.39)/93.97 |
| MAHALANOBIS (REN ET AL., 2021) | 81.26(0.53) | **78.05**(0.09) | — | 91.34(0.62) | 89.78(0.54) |
| MOS (HUANG & LI, 2021) | **82.41**(0.02) | 70.00(0.72) | — | **96.66**(0.23) | **93.66**(0.22) |
| HSC ($\alpha = 1$, $\beta = 0$, PRED) | 82.38(0.06)/83.25 | 76.78(3.38)/79.80 | — | 93.93(0.22)/95.08 | 91.33(0.28)/92.38 |
| HSC ($\alpha = 1$, $\beta = 0$, $H_{\text{mean}}$) | 82.38(0.06)/83.25 | 77.27(3.83)/75.86 | — | 96.90(0.11)/96.89 | 93.92(0.20)/93.17 |
| HSC ($\alpha = 1$, $\beta = 0$, $H_{\text{max}}$) | 82.38(0.06)/83.25 | 76.11(3.02)/75.21 | — | 91.50(0.40)/94.81 | 89.16(0.46)/91.35 |
| HSC ($\alpha = 1$, $\beta = 0$, $H_{\text{min}}$) | 82.38(0.06)/83.25 | 69.10(6.96)/72.92 | — | 98.04(0.07)/98.26 | 93.70(0.13)/93.98 |
| HSC ($\alpha = 1$, $\beta = 0$, ENERGY) | 82.38(0.06)/83.25 | 75.28(1.01)/71.28 | — | **98.22**(0.11)/97.96 | 94.17(0.15)/93.25 |
| HSC ($\alpha = 1$, $\beta = 0.2$, PRED) | **82.85**(0.14)/**84.05** | **79.40**(0.76)/**80.67** | — | 95.06(0.13)/96.05 | 92.29(0.15)/93.33 |
| HSC ($\alpha = 1$, $\beta = 0.2$, $H_{\text{mean}}$) | **82.85**(0.14)/**84.05** | **79.40**(0.67)/76.35 | — | 97.23(0.11)/96.93 | 94.08(0.13)/93.30 |
| HSC ($\alpha = 1$, $\beta = 0.2$, $H_{\text{max}}$) | **82.85**(0.14)/**84.05** | 78.81(0.68)/76.02 | — | 94.57(0.29)/95.49 | 91.79(0.27)/92.05 |
| HSC ($\alpha = 1$, $\beta = 0.2$, $H_{\text{min}}$) | **82.85**(0.14)/**84.05** | 76.81(0.73)/78.00 | — | 98.15(0.07)/98.49 | **94.38**(0.07)/**94.87** |
| HSC ($\alpha = 1$, $\beta = 0.2$, ENERGY) | **82.85**(0.14)/**84.05** | 75.29(1.18)/71.32 | — | **98.22**(0.06)/**98.51** | 94.17(0.17)/93.71 |
| IMAGENET 1K | | | | | |
| MSP (HENDRYCKS & GIMPEL, 2017) | 74.94(0.08)/**77.05** | 74.30(0.24)/74.90 | 79.33(0.17)/81.32 | 80.42(0.19)/82.71 | 77.96(0.11)/79.57 |
| ODIN (LIANG ET AL., 2018) | 74.94(0.08)/**77.05** | **76.25**(0.11)/77.62 | **79.84**(0.21)/**81.82** | 81.95(0.15)/84.02 | **79.18**(0.13)/**80.98** |
| ENERGY (LIANG ET AL., 2018) | 74.94(0.08)/**77.05** | 76.20(0.08)/**77.83** | 79.28(0.21)/81.05 | 81.45(0.17)/83.24 | 78.80(0.13)/80.53 |
| MOS (HUANG & LI, 2021) | **75.00**(0.43) | 74.71(0.90) | 74.00(0.53) | **87.11**(0.40) | 77.32(0.59) |
| HSC ($\alpha = 1$, $\beta = 0$, PRED) | 73.79(0.13)/76.51 | 72.73(0.47)/73.42 | 78.33(0.27)/80.49 | 80.64(0.17)/82.92 | 77.07(0.24)/78.78 |
| HSC ($\alpha = 1$, $\beta = 0$, $H_{\text{mean}}$) | 73.79(0.13)/76.51 | 64.84(0.64)/61.45 | 77.03(0.28)/75.02 | 82.86(0.11)/85.43 | 74.47(0.31)/73.09 |
| HSC ($\alpha = 1$, $\beta = 0$, $H_{\text{max}}$) | 73.79(0.13)/76.51 | 73.65(0.35)/70.25 | 77.19(0.24)/77.57 | 77.01(0.18)/78.69 | 76.00(0.22)/75.47 |
| HSC ($\alpha = 1$, $\beta = 0$, $H_{\text{min}}$) | 73.79(0.13)/76.51 | 52.97(2.35)/59.65 | 57.05(0.57)/65.10 | 69.41(0.21)/73.53 | 58.64(0.46)/65.33 |
| HSC ($\alpha = 1$, $\beta = 0$, ENERGY) | 73.79(0.13)/76.51 | 54.36(1.52)/51.40 | 78.12(0.15)/76.77 | 86.27(0.09)/88.07 | 72.36(0.54)/71.24 |
| HSC ($\alpha = 1$, $\beta = 0.05$, PRED) | 74.46(0.06)/**76.79** | 72.86(0.56)/**73.69** | **79.40**(0.29)/**81.45** | 82.38(0.58)/84.35 | **77.99**(0.41)/**79.63** |
| HSC ($\alpha = 1$, $\beta = 0.05$, $H_{\text{mean}}$) | 74.46(0.06)/**76.79** | 63.54(0.74)/61.56 | 76.62(0.35)/75.44 | 84.54(0.40)/86.01 | 74.27(0.34)/73.45 |
| HSC ($\alpha = 1$, $\beta = 0.05$, $H_{\text{max}}$) | 74.46(0.06)/**76.79** | **73.83**(0.37)/70.24 | 77.93(0.04)/78.37 | 78.87(0.20)/78.89 | 76.83(0.18)/75.87 |
| HSC ($\alpha = 1$, $\beta = 0.05$, $H_{\text{min}}$) | 74.46(0.06)/**76.79** | 59.13(1.38)/62.92 | 58.96(0.51)/62.16 | 69.49(1.51)/74.22 | 61.50(0.91)/65.25 |
| HSC ($\alpha = 1$, $\beta = 0.05$, ENERGY) | 74.46(0.06)/**76.79** | 57.46(1.11)/54.65 | 78.05(0.71)/77.88 | **88.13**(0.11)/**89.81** | 73.77(0.59)/73.19 |

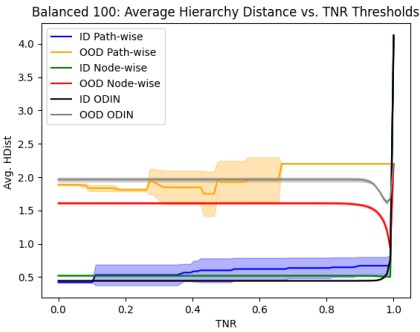

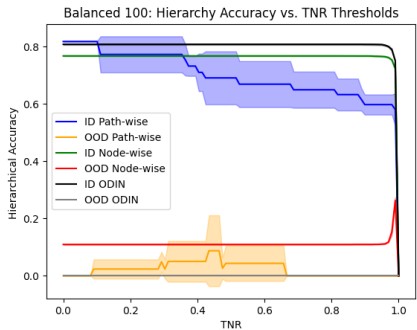

Figure 6: Balanced Imagenet 100 average hierarchy distance vs. TNR threshold values.

Figure 7: Balanced Imagenet 100 hieararchy accuracy vs. TNR threshold values.

A.8   IMAGENET 1K HIERARCHY DISTANCE AND ACCURACY

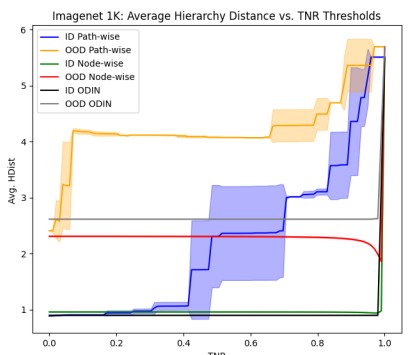

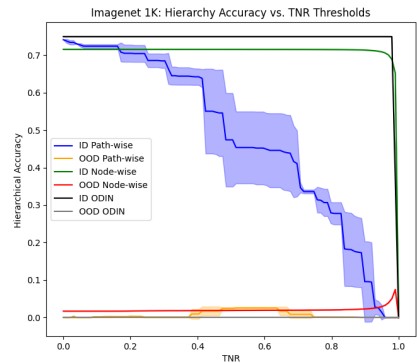

Figure 8: Imagenet 1K average hierarchy distance vs. TNR threshold values.

Figure 9: Imagenet 1K hieararchy accuracy vs. TNR threshold values.

A.9   ID AND OOD INFERENCE ACCURACY VS. TNR

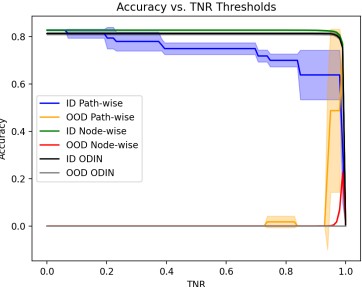

Figure 10: Imagenet-100 ID and OOD accuracy across TNR threshold values for path-wise and synset-wise threshold metrics with ODIN baseline.

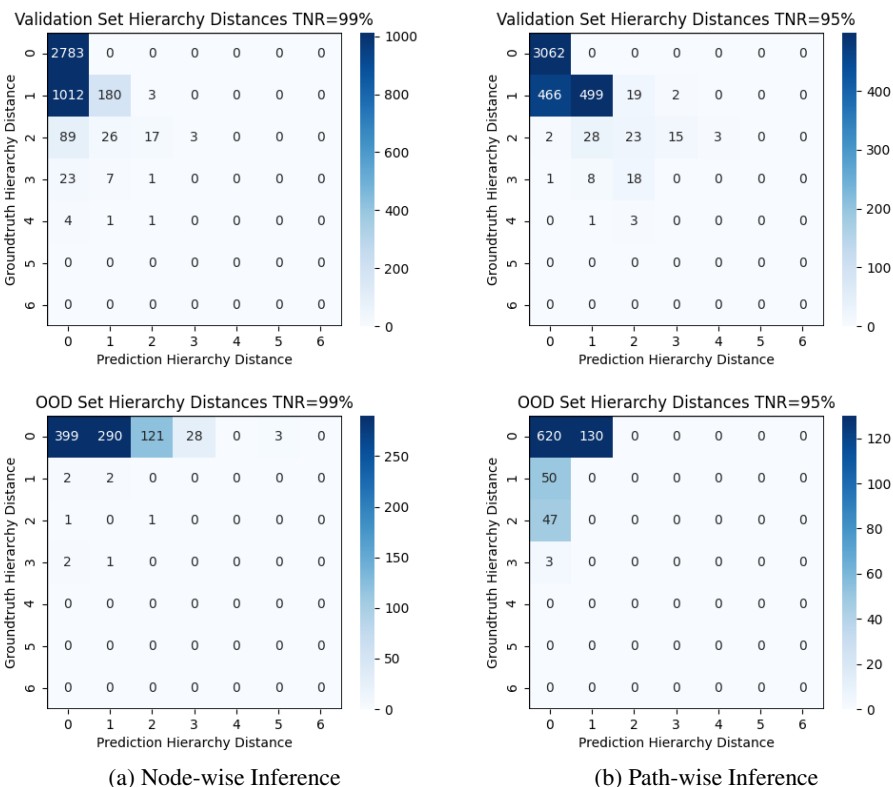

(a) Node-wise Inference        (b) Path-wise Inference

Figure 11: Imagenet-100 path- and node-wise inference hierarchy distance confusion matrices on ID and OOD data.

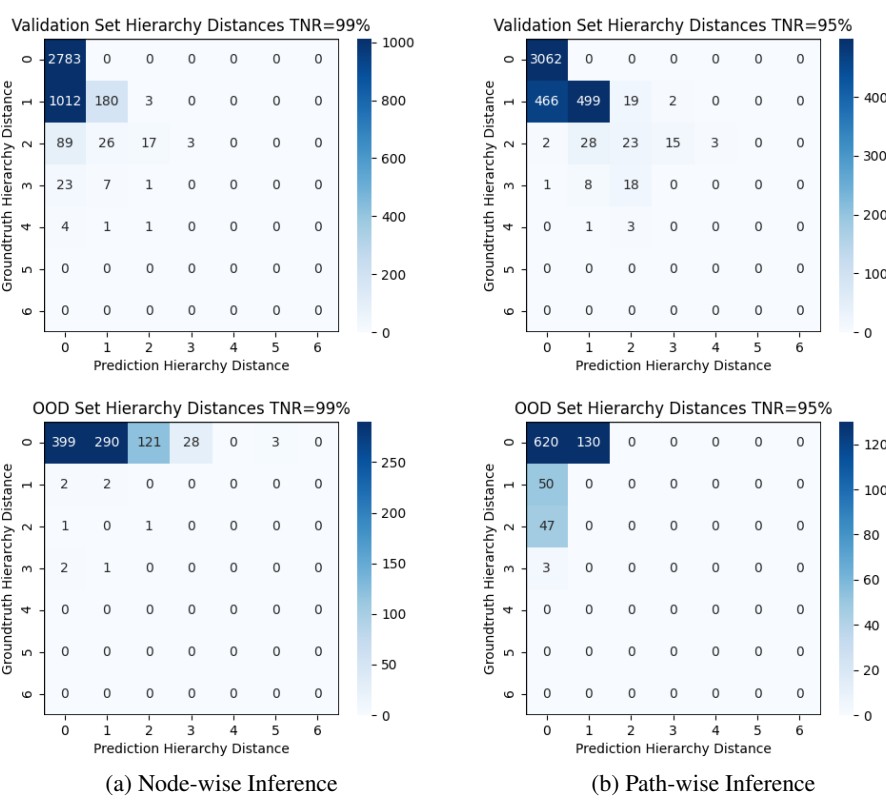

(a) Node-wise Inference          (b) Path-wise Inference

Figure 12: Imagenet-100 path- and node-wise inference hierarchy distance confusion matrices on ID and OOD data.

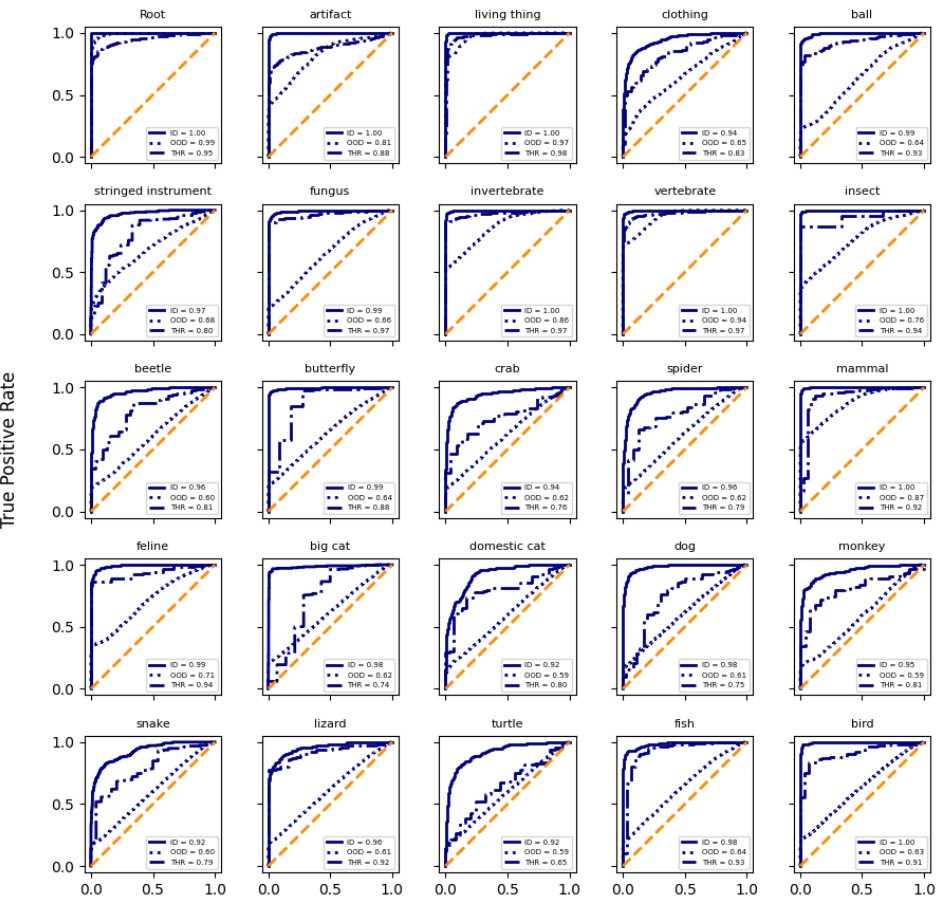

Figure 13: Imagenet-100 synset micro-ROC curves for ID data only, ID and OOD, and ID and OOD with a TNR=0.95 prediction path threshold.

## A.13 Hierarchical performance by depth

Hierarchical classifiers provide interpretable decisions by introducing additional classification tasks at each internal node in the hierarchy. While this allows for improved interpretability and ability to perform verification, each additional classifier is another potential point of failure. We mitigate this by using "soft" decisions by using path probabilities for predictions instead of traversing the hierarchy in a top-down fashion akin to a decision tree as suggested by Wan et al. (2021). We plot percentage of classification errors that occur at each hierarchy depth (figs. 14 to 16) and compare the performance of hierarchical classifiers to a flat classifier. To determine the depth of the error we follow the prediction path in a top-down fashion and record where the depth of the node where the error occurs. For example, for the Imagenet 100 dataset, if a sample's groundtruth is "house finch" and the classifier predicts "puffer", then the error was made by the "vertebrate" node's classifier and the error depth is 2 (see fig. 2). Note that this can be performed for any prediction regardless of whether the classifier can only produce leaf node predictions. We calculate OOD error depths by assigning each OOD sample a ground truth to be the deepest internal node that is contained in the set of ID classes as we do when calculating hierarchical accuracy and hierarchical distance (section 5.2). As we are trying to evaluate the prediction accuracy and not the inference mechanism, we calculate an "oracle" accuracy for these plots. Specifically, an OOD prediction is "correct" if its path contains the OOD groundtruth path and predicted nodes that are at deeper levels than the groundtruth are ignored. For example, in the Imagenet 100 dataset "junco" is OOD and its groundtruth value is assigned to its closest ancestor that is contained in the ID set which is "bird". For the purposes of these plots, if a classifier predicts "house finch" whose path also contains "bird", the prediction is considered to be correct (see fig. 2).

In figs. 14 to 16 it is clear that hierarchical and flat classifiers tend to misclassify similar difficult samples and have similar ID and OOD accuracy performance. Also, note that when the OOD accuracy performance drops in the Imagenet 1K dataset (fig. 16), the hierarchical classifiers begin to outperform the flat classifier. Since we keep the feature extractor the same between hierarchical and flat classifiers it is perhaps not surprising that they learn similar feature sets that lead to similar leaf node mistakes. The main focus of the current work is to compare the OOD performance of hierarchical versus flat models. This restricts our ability to explore additional hierarchical feature extractors as it would not allow for direct performance comparisons. An extension of the current work would explore hierarchical feature extractors to improve each node's classification performance by learning additional features that are optimized to distinguish between a given node's children. Another interesting future direction is to identify poorly performing internal nodes' classifiers during training and explicitly attempt to improve their performance through adapting the nodes' weights (eq. (3)) or by developing other techniques.

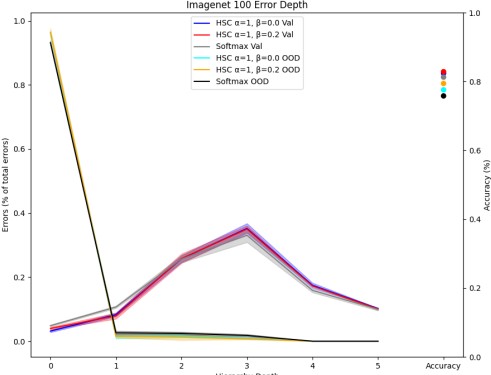

Figure 14: Imagenet 100 errors by hierarchy depth for ID and OOD datasets. The mean accuracy is given in the right-most column and its scale is on the right-hand side.

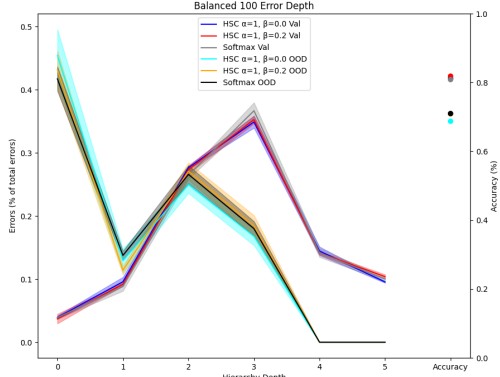

Figure 15: Balanced Imagenet 100 errors by hierarchy depth for ID and OOD datasets. The mean accuracy is given in the right-most column and its scale is on the right-hand side.

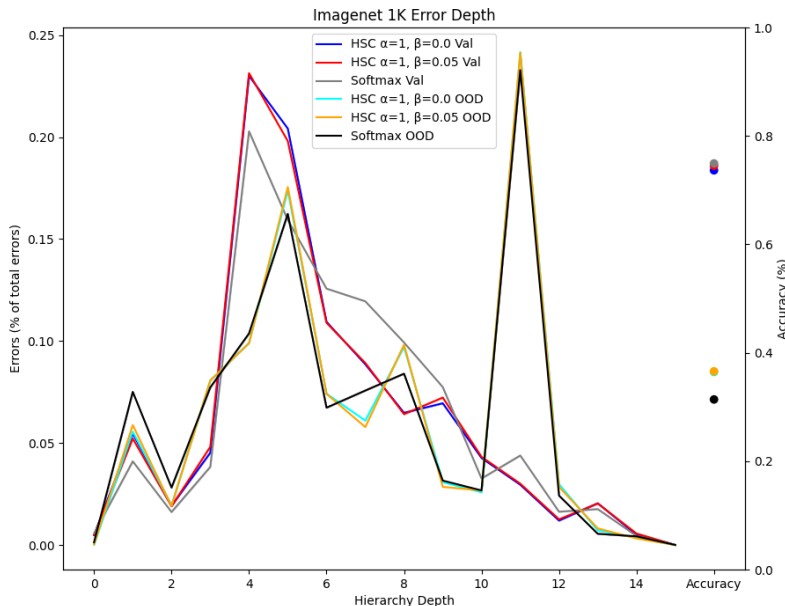

Figure 16: Imagenet 1K errors by hierarchy depth for ID and OOD datasets. The mean accuracy is given in the right-most column and its scale is on the right-hand side.

## A.14 FULLY CONNECTED HEAD EXPERIMENTS

Table 7: Hierarchical softmax classifier (HSC) performance on the Imagenet-100 dataset when adding additions fully-connected (FC) layers to classification head. AUROC scores are provided for each OOD method: MSP (Hendrycks & Gimpel, 2017), ODIN (Liang et al., 2018), MOS (Wan et al., 2021). Note that the node-wise scaling of the HSC methods is different from table 6 causing the discrepancy in HSC numbers. All numbers are percentages.

| MODEL | ACCURACY | BASELINE AUROC | PATH PREDICTION | PATH ENTROPY | |
| | | | | MEAN | MIN |
| --- | --- | --- | --- | --- | --- |
| IMAGENET 100 | | | | | |
| MSP | 81.26(0.53) | 90.25(0.62) | — | — | — |
| ODIN | 81.26(0.53) | 93.20(0.36) | — | — | — |
| MOS | 82.41(0.02) | 93.66(0.22) | — | — | — |
| MSP FC3 | 82.12(0.29) | 90.68(0.41) | — | — | — |
| ODIN FC3 | 82.12(0.29) | 93.78(0.31) | — | — | — |
| MOS FC3 | 81.82(0.15) | 93.49(0.49) | — | — | — |
| HSC ($\alpha = 1, \beta = 0$) | 78.36(0.87) | — | 89.09(0.65) | 92.39(0.43) | 92.68(0.24) |
| HSC ($\alpha = 1, \beta = 0.2$) | 83.05(0.12) | — | 92.16(0.26) | 93.93(0.27) | 94.29(0.27) |
| HSC FC3 ($\alpha = 1, \beta = 0$) | 81.90(0.08) | — | 91.51(0.35) | 93.86(0.20) | 93.46(0.18) |
| HSC FC3 ($\alpha = 1, \beta = 0.2$) | 82.73(0.24) | — | 91.80(0.46) | 93.78(0.28) | 94.57(0.20) |

## A.15 SUPPLEMENTAL OOD GRANULARITY PERFORMANCE

Table 8: Hierarchical classifier performance on the fine-grain Imagenet-1K datasets. AUROC scores are provided for each OOD method: (Ours) Hierarchical softmax classifier (HSC), MSP (Hendrycks & Gimpel, 2017), ODIN (Liang et al., 2018), MOS (Wan et al., 2021). All models are ResNet50 architectures trained for 90 epochs. FC: Number of fully-connected layers for classifier

| DATASET | MODEL | AUROC | | |
| --- | --- | --- | --- | --- |
| | | FINE | MEDIUM | COARSE |
| IMAGENET 100 | MSP | $72.47_{(0.31)}$ | — | $92.62_{(0.67)}$ |
| | ODIN | $72.93_{(1.87)}$ | — | $95.90_{(0.47)}$ |
| | MOS | $70.00_{(0.72)}$ | — | $96.66_{(0.23)}$ |
| | MSP FC3 | $72.55_{(1.22)}$ | — | $93.09_{(0.39)}$ |
| | ODIN FC3 | $69.05_{(0.75)}$ | — | $97.08_{(0.26)}$ |
| | MOS FC3 | $69.78_{(2.65)}$ | — | $96.66_{(0.21)}$ |
| | HSC (PRED) | $74.63_{(0.92)}$ | — | $91.02_{(0.67)}$ |
| | HSC ($H_{\text{mean}}$) | $73.09_{(0.95)}$ | — | $94.96_{(0.46)}$ |
| | HSC ($H_{\text{min}}$) | $62.24_{(0.72)}$ | — | $96.74_{(0.25)}$ |
| | HSC OE (PRED) | $71.11_{(1.98)}$ | — | $94.96_{(0.04)}$ |
| | HSC OE ($H_{\text{mean}}$) | $70.01_{(1.93)}$ | — | $97.11_{(0.07)}$ |
| | HSC OE ($H_{\text{min}}$) | $65.14_{(1.40)}$ | — | $98.18_{(0.13)}$ |
| | HSC FC3 (PRED) | $72.22_{(0.56)}$ | — | $94.09_{(0.44)}$ |
| | HSC FC3 ($H_{\text{mean}}$) | $71.55_{(0.49)}$ | — | $96.83_{(0.28)}$ |
| | HSC FC3 ($H_{\text{min}}$) | $59.69_{(1.57)}$ | — | $97.96_{(0.01)}$ |
| | HSC FC3 OE (PRED) | $71.86_{(0.17)}$ | — | $94.46_{(0.53)}$ |
| | HSC FC3 OE ($H_{\text{mean}}$) | $70.67_{(0.25)}$ | — | $96.86_{(0.33)}$ |
| | HSC FC3 OE ($H_{\text{min}}$) | $68.87_{(0.11)}$ | — | $97.99_{(0.23)}$ |
| IMAGENET 1K | MSP | $74.30_{(0.24)}$ | $79.33_{(0.17)}$ | $80.42_{(0.19)}$ |
| | ODIN | $76.25_{(0.11)}$ | $79.84_{(0.21)}$ | $81.95_{(0.15)}$ |
| | MOS | $74.71_{(0.90)}$ | $74.00_{(0.53)}$ | $87.11_{(0.40)}$ |
| | HSC (PRED) | $72.73_{(0.47)}$ | $78.33_{(0.27)}$ | $80.64_{(0.17)}$ |
| | HSC ($H_{\text{mean}}$) | $64.84_{(0.64)}$ | $77.03_{(0.28)}$ | $82.86_{(0.11)}$ |
| | HSC ($H_{\text{min}}$) | $52.97_{(2.35)}$ | $57.05_{(0.57)}$ | $69.41_{(0.21)}$ |
| | HSC OE (PRED) | $72.62_{(0.20)}$ | $79.19_{(0.19)}$ | $82.00_{(0.44)}$ |
| | HSC OE ($H_{\text{mean}}$) | $63.65_{(0.60)}$ | $76.60_{(0.49)}$ | $84.23_{(0.23)}$ |
| | HSC OE ($H_{\text{min}}$) | $60.64_{(0.22)}$ | $59.02_{(0.57)}$ | $69.60_{(1.81)}$ |

## A.16 PERFORMANCE ON FAR-OOD BENCHMARKS

Table 9: Coarse-grain OOD dataset baseline performance. AUROC scores are provided for each OOD method: (Ours) Hierarchical softmax classifier (HSC), MSP (Hendrycks & Gimpel, 2017), ODIN (Liang et al., 2018), MOS (Wan et al., 2021). All models are ResNet50 architectures trained for 90 epochs. FC: Number of fully-connected layers for classifier

| ID Dataset | Model | AUROC | | | |
|---|---|---|---|---|---|
| | | iNaturalist | SUN | Places | Textures |
| IMAGENET 100 | MSP | $92.22_{(0.46)}$ | $93.62_{(0.23)}$ | $92.24_{(0.17)}$ | $88.60_{(0.57)}$ |
| | ODIN | $95.60_{(0.34)}$ | $97.30_{(0.19)}$ | $96.10_{(0.13)}$ | $94.85_{(0.35)}$ |
| | MOS | $93.50_{(0.04)}$ | $95.85_{(0.04)}$ | $94.72_{(0.12)}$ | $95.13_{(0.21)}$ |
| | MSP FC3 | $93.04_{(0.23)}$ | $94.77_{(0.03)}$ | $93.43_{(0.13)}$ | $90.07_{(0.13)}$ |
| | ODIN FC3 | $96.52_{(0.10)}$ | $98.00_{(0.07)}$ | $96.87_{(0.04)}$ | $95.89_{(0.12)}$ |
| | MOS FC3 | $93.83_{(0.32)}$ | $95.89_{(0.20)}$ | $94.83_{(0.14)}$ | $94.78_{(0.18)}$ |
| | HSC (PRED) | $91.22_{(0.27)}$ | $92.64_{(0.59)}$ | $91.25_{(0.76)}$ | $87.69_{(0.03)}$ |
| | HSC ($H_{mean}$) | $91.83_{(0.26)}$ | $93.40_{(0.55)}$ | $92.33_{(0.71)}$ | $89.83_{(0.04)}$ |
| | HSC ($H_{min}$) | $86.50_{(0.82)}$ | $94.19_{(0.68)}$ | $92.66_{(0.87)}$ | $92.88_{(0.06)}$ |
| | HSC OE (PRED) | $94.05_{(0.40)}$ | $95.73_{(0.19)}$ | $94.59_{(0.03)}$ | $91.51_{(0.15)}$ |
| | HSC OE ($H_{mean}$) | $94.38_{(0.29)}$ | $96.31_{(0.18)}$ | $95.45_{(0.02)}$ | $93.44_{(0.14)}$ |
| | HSC OE ($H_{min}$) | $90.73_{(0.39)}$ | $96.30_{(0.19)}$ | $95.06_{(0.19)}$ | $95.22_{(0.22)}$ |
| | HSC FC3 (PRED) | $93.25_{(0.48)}$ | $95.25_{(0.36)}$ | $93.80_{(0.36)}$ | $90.82_{(0.21)}$ |
| | HSC FC3 ($H_{mean}$) | $94.09_{(0.47)}$ | $96.08_{(0.29)}$ | $94.87_{(0.28)}$ | $92.79_{(0.17)}$ |
| | HSC FC3 ($H_{min}$) | $91.93_{(0.47)}$ | $96.79_{(0.12)}$ | $95.53_{(0.10)}$ | $95.29_{(0.19)}$ |
| | HSC FC3 OE (PRED) | $93.83_{(0.20)}$ | $95.68_{(0.12)}$ | $94.32_{(0.18)}$ | $90.82_{(0.29)}$ |
| | HSC FC3 OE ($H_{mean}$) | $94.13_{(0.10)}$ | $96.19_{(0.14)}$ | $95.16_{(0.12)}$ | $92.69_{(0.22)}$ |
| | HSC FC3 OE ($H_{min}$) | $91.20_{(0.76)}$ | $96.42_{(0.14)}$ | $95.14_{(0.11)}$ | $94.93_{(0.06)}$ |
| IMAGENET 1K | MSP | $88.16_{(0.13)}$ | $81.05_{(0.16)}$ | $80.63_{(0.18)}$ | $80.18_{(0.23)}$ |
| | ODIN | $91.05_{(0.33)}$ | $86.12_{(0.39)}$ | $84.72_{(0.31)}$ | $85.40_{(0.59)}$ |
| | MOS | $95.22_{(0.35)}$ | $92.06_{(0.14)}$ | $90.59_{(0.13)}$ | $83.32_{(0.98)}$ |
| | HSC (PRED) | $88.56_{(0.50)}$ | $80.77_{(0.30)}$ | $80.36_{(0.15)}$ | $80.43_{(0.23)}$ |
| | HSC ($H_{mean}$) | $89.92_{(0.38)}$ | $86.08_{(0.22)}$ | $84.75_{(0.11)}$ | $82.93_{(0.23)}$ |
| | HSC ($H_{min}$) | $80.04_{(0.86)}$ | $88.22_{(0.55)}$ | $85.66_{(0.48)}$ | $75.77_{(0.38)}$ |
| | HSC OE (PRED) | $88.03_{(0.15)}$ | $80.31_{(0.06)}$ | $79.90_{(0.23)}$ | $80.31_{(0.44)}$ |
| | HSC OE ($H_{mean}$) | $88.21_{(0.45)}$ | $85.57_{(0.03)}$ | $84.17_{(0.15)}$ | $82.51_{(0.33)}$ |
| | HSC OE ($H_{min}$) | $72.39_{(3.60)}$ | $78.21_{(0.92)}$ | $76.68_{(0.94)}$ | $78.75_{(0.97)}$ |

