# OpenReview forum: "Fine-grain Inference on Out-of-Distribution Data with Hierarchical Classification"
_ICLR.cc/2023/Conference — Submitted to ICLR 2023_

### Official Review · Reviewer_NPqs · 2022-10-22

**Confidence:** 2
**Correctness:** 4
**Technical Novelty And Significance:** 2
**Empirical Novelty And Significance:** 2
**Recommendation:** 3

**Clarity, Quality, Novelty And Reproducibility:**

Clarity, Quality, Novelty, Reproducibility

* The paper is mostly clear. I don't like the OOD terminology, but that appears to be standard.
* The proposed methodology and results do not appear to be particularly novel. Various hierarchical losses have been proposed before. The novelty/benefit of this particular loss should be better highlighted.
* The numerical results don't seem to be huge improvements.
* Seems sufficiently reproducible.


**Strength And Weaknesses:**

Strengths:

* Seems like a reasonable approach.


Weaknesses

* The stopping condition, Section 3.4, merits further discussion on terminology (also for Section 5.1, 5.3).
* Additionally, a histogram showing the distribution of entropy values for in distribution data (ID) and OOD might be informative on how well one might expect to determine the threshold.


Question

* Not really a strength or weakness, but the hypothesis seems to be that if a OOD sample is given, then at some node on the correct classification path each child should be equally likely (eqn 4). Is this really reasonable? Does this really appear to be the case when OOD (and ID) errors? Section 5.3 might be a place to expand the analysis.



**Summary Of The Paper:**

Summary:

This paper examines multi-class classification in settings with out-of-distribution (OOD) data. Here, this refers to test data where the labels are not available at training time. The proposed solution is hierarchical classification, with the leaf nodes of the tree being the classes seen in training. The tree is given, so intermediate nodes may be interpretable. For classes not seen in training, the goal is to classify as deep within the tree as possible and ultimately conclude that the sample does not match any seen labels. To achieve these goals, training is done with the proposed hierarchical OOD loss. A decision to stop classifying deeper into the tree is made by a threshold on prediction path entripy measures. Experiments are conducted on a fine-grained OOD dataset and a coarse-grained OOD dataset. The difference is that fine-grained OOD datasets may exhibit deep classifications within the tree even for OOD samples, but coarse-grained OOD datasets mostly contain OOD decisions at a shallow level. Numerical performance appears reasonable.


**Summary Of The Review:**

Summary

* There are some potentially useful, if not too interesting, ideas.
* In any case, the terminology and analysis should be improved.

---

> ### Author Response · Authors · 2022-11-14
> **Response**
>
> We thank the reviewer for their comments. The major contribution of our work is leveraging hierarchical classification to improve the interpretability and utility of the model under OOD data. We highlight the fine-, medium-, and coarse-grain OOD scenario as this is a more realistic scenario for many real-world applications when compared to far OOD problems which may include non-semantic factors that can be utilized by the classifier to separate OOD samples. We provide extensive experiments on large-scale, granular ID/OOD problem sets and show how hierarchical techniques provide more interpretable predictions that are impossible to achieve with flat classification while maintaining standard OOD AUROC performance. Finally, we are suggesting a paradigm shift for the field of OOD detection: instead of focusing on incremental improvements in AUROC scores for the binary ID/OOD detector, new methods should focus on improving interpretability and providing more information to the user via supporting non-leaf node inference under uncertainty.
>
> > Additionally, a histogram showing the distribution of entropy values for in distribution data (ID) and OOD might be informative on how well one might expect to determine the threshold.
>
> We provide the micro-averaged ROC curves for each (internal) node in the hierarchy (Figure 13). The ROCs reflect how separable the ID and OOD samples are and the tradeoff between TPR and FPR used to select the threshold.
>
> > Not really a strength or weakness, but the hypothesis seems to be that if a OOD sample is given, then at some node on the correct classification path each child should be equally likely (eqn 4). Is this really reasonable?
>
> We agree that the hypothesis that the each ID class should be equally likely is not reasonable as the OOD sample may be more similar to one of the ID classes. However, it is not necessary that all of the children are "equally" likely, rather there should be less certainty that the sample is one particular class compared to training set samples. Since the OOD sample does not perfectly fit any of the ID classes there should be more uncertainty when compared to ID samples. We propose the $\mathcal{L}_\mathsf{other}$ loss to encourage classes that are outside of the node's scope to have less certainty by driving them towards a uniform distribution. Note that this also acts to calibrate each of the node's classifiers as mentioned in reviewer Wf9S's review.

---

### Official Review · Reviewer_KPxY · 2022-11-01

**Confidence:** 4
**Correctness:** 4
**Technical Novelty And Significance:** 3
**Empirical Novelty And Significance:** 3
**Recommendation:** 8

**Clarity, Quality, Novelty And Reproducibility:**

The paper is well-written, clearly explaining the methods and experiments. The novelty of the work lies in the framework designed towards explainability aspect of OOD detection. I would strongly recommend the authors to open-source the code for reproducibility.

**Strength And Weaknesses:**

Positives:

- I completely agree with the authors that such an approach is highly interpretable and this is a very relevant research direction in AI safety.
- Also, providing a less specific correct coarse prediction is more useful than an incorrect highly specific prediction, which the method provides. In practice most encountered OOD samples would be such near-OOD samples where such a response is preferred over just calling out OOD samples.
- The paper provides a very interesting set of experiments over fine/ coarse grained OOD prediction and analyzes the result.
- Very exhaustive comparison with existing methods.

Potential improvements:

- One key advantage of this approach is the explainability/ interpretability. Some metrics on this would be useful. For example: In depth case study on "partial OOD" samples (which is OOD in fine-grain but ID in mid-level of hierarchy) and mistakes on detecting this would be interesting.
- Similarly, some downstream cost analysis (penalizing different kinds of mistakes at different hierarchies) could reflect how such a framework would be useful in the real world.
- Showing some potential failure modes and discussion on them would be very useful for the community, especially to pave the direction for future work.

**Summary Of The Paper:**

 In this paper, the authors propose a hierarchical classification framework for both fine-grained and coarse-grained OOD detection. This involves creating a tree based structure from the set of classification labels to establish a hierarchy. Once established, the authors introduce a new loss function to train the model maintaining this hierarchy. The authors also propose a new OOD detection score based on prediction path entropy.

I have reviewed this paper before and the authors have addressed most of my previous concerns.


**Summary Of The Review:**

I would recommend this paper be accepted. The AI safety community has focussed on developing OOD detection methods in a binary sense accept/ reject. Most papers develop methods which improves the OOD-AUROC marginally. We are at a state where just 1-2 points improvement using a new method won't have any practical value. This paper looks beyond the traditional binary treatment of OOD detection. This is very useful in real world deployment where just showing OOD without explanation was degrade user trust and lead to a bad experience overall.

---

> ### Author Response · Authors · 2022-11-14
> **Response**
>
> We thank the reviewer for the positive feedback. We are glad that they appreciate the importance of developing OOD methods that go beyond the standard AUROC scores. We also appreciate the comments highlighting opportunities for improving our analysis. We agree that an in-depth analysis of inference performance across granularities of OOD and failure modes will help identify research directions and we have provided additional depth-wise accuracy tables for ID and OOD samples in the appendix. A comprehensive failure mode analysis is an interesting area for future works. Finally, we will open-source the dataset splits, hierarchies, results, and code to improve openness, reproducibility, and accelerate future research into hierarchical OOD methods.

---

### Official Review · Reviewer_XEo4 · 2022-11-01

**Confidence:** 2
**Correctness:** 3
**Technical Novelty And Significance:** 2
**Empirical Novelty And Significance:** 1
**Recommendation:** 6

**Clarity, Quality, Novelty And Reproducibility:**

There are several places where the thread of the argument is not clear-- such as what would be a clarity test for a sample being OOD.  How could one construct an oracle?  What would it mean to say that a case does not belong to the population, as opposed to the concepts covered by the training sample? Similarly the paper speaks of "otherness." How is this defined?

The paper seems both to be concerned with the central ID / OOD question (which lacks some clarity) and the loosely related question of the performance of flat versus hierarchical classifiers.

**Strength And Weaknesses:**

The idea of learning a hierarchical classifier is valuable, in terms of giving better insights into the results, scaling to larger numbers of classes and assigning a more appropriate level of granularity to the classification.

In concept the classifier's marginal probability ascribed to the image -- the probability that it would arise in the population, irrespective of it's class should give a measure of ID versus OOD.  This would be a principled alternative to the measures used in the paper.



**Summary Of The Paper:**

The paper applies a hierarchical supervised classifier on which a measure whether an observed image should be considered "in-distribution" or "out of distribution."  This considered both in quality -- how specific ("fine-grained") the classification is -- and degree, by at what point in the hierarchy to consider a sample "OOD."

**Summary Of The Review:**

Based on a cursory understanding of the work, it appears to have value, but it's hard for me to put a finger on exactly the novelty and soundness of the work.  It does illustrate some interesting characteristics of hierarchical image classification, but comes to no clear conclusions.

---

> ### Author Response · Authors · 2022-11-14
> **Response**
>
> Thank you for your comments.
>
> >In concept the classifier's marginal probability ascribed to the image -- the probability that it would arise in the population, irrespective of it's class should give a measure of ID versus OOD. This would be a principled alternative to the measures used in the paper.
>
> We agree that density estimation is an interesting problem in the context of
> OOD detection. Specifically, flow based models, VAEs, and GANs are
> interesting topics of ongoing research as well as hybrid modeling techniques.
> Currently, density estimation techniques have underperformed on OOD tasks [1,2].
> Another OOD approach that utilizes an energy-based model (Liu et al 2020)
> estimates the probability density p(x) with energy values using the Gibbs
> distribution. We have added the energy score to our baseline values in the updated submission (Tables 3 & 6).
> The energy score slightly underperforms the ODIN flat classifier baseline.
> Hierarchical classifiers using the energy score underperform on the Imagenet-1K and Imagenet-100 datasets and outperform our method on the Balanced-100 dataset. The energy score does not consistently improve OOD detection in our experiments.
>
> > There are several places where the thread of the argument is not clear-- such as what would be a clarity test for a sample being OOD. How could one construct an oracle? What would it mean to say that a case does not belong to the population, as opposed to the concepts covered by the training sample? Similarly the paper speaks of "otherness." How is this defined?
>
> Current OOD detection literature determines whether a sample is ID or OOD based
> on whether the image depicts a target class for which the model was trained.
> Historically, the OOD dataset is chosen from a different dataset that does not
> resemble the ID set (Hendrycks & Gimpel, 2017). For example, comparing CIFAR-10 images to MNIST digits, Gaussian noise or SUN database.
> More recently nearer, more difficult OOD sets have been of interest.
> For example, CIFAR-10 versus CIFAR-100 which are the same modality, but contain
> disjoint class sets (cats in CIFAR-10 and fish in CIFAR-100).
> In this work, we expand upon previous works by measuring the
> ability of OOD techniques to distinguish between ID and OOD across granularity
> levels as well as make predictions on OOD samples that are close to the ID set.
> Concretely, given a set of leaf nodes $\mathcal{Y}$ (e.g. Imagenet-1K classes)
> we select a subset of these classes to be the ID set $\mathcal{Y}_\mathsf{ID}
> \subset \mathcal{Y}$ and all other classes to be the OOD set
> $\mathcal{Y}_\mathsf{OOD} = \mathcal{Y} \setminus \mathcal{Y}_\mathsf{ID}$.
> ID training and validation sets are be created from the ID set (
> $\mathcal{D}_\mathsf{ID} = \{(x_i, y_i) : y_i \in \mathcal{Y}_\mathsf{ID}\}$)
> and likewise for the OOD dataset (
> $\mathcal{D}_\mathsf{OOD} = \{(x_i, y_i) : y_i \in \mathcal{Y}_\mathsf{OOD}\}$).
> An oracle for the ID accuracy and ID vs. OOD AUROC metrics would perform the following operation:
>
> \begin{eqnarray}
> (x,y)  \\sim \\mathcal{D} = \\{(x_i,y_i):y_i \\in \\mathcal{Y}\\} \\\\
>   f_\mathsf{oracle} : x \mapsto
>   \begin{cases}
>     y                 & \text{if } y \in \mathcal{Y}_\mathsf{ID} \\\\
>     \text{OOD} & \text{otherwise}
>   \end{cases}
> \end{eqnarray}
>
> As the OOD class set becomes more similar to the ID class set, the distinction
> between ID and OOD is less clear and a classifier trained on the ID set will
> have learned some information regarding the OOD samples. This is our motivation
> for the current work. We aim to provide more information to the user by
> predicting internal nodes when the uncertainty over the leaf nodes is too high.
>
> An oracle for the hierarchical classifier would output the deepest node of the
> hierarchy that contains the node or one of its ancestors:
> \begin{align}
>   (x,y,n) \sim \mathcal{D} &= \{
>     (x_i,y_i,n_i): y_i \in \mathcal{Y},
>     n_i = \max(\mathsf{anc}(y_i) \cap  \mathcal{H})
>   \} \\\\
>   f_\mathsf{oracle} : x &\mapsto n
> \end{align}
> Note that $\mathsf{anc}(y)$ includes $y$, the hierarchy $\mathcal{H}$
> only includes leaf nodes in $\mathcal{Y}_\mathsf{ID}$ and the respective internal nodes, and $n$ increases with depth so that the deepest ancestor is $\max(n) \in \mathsf{anc}(y)$.
>
> When creating the $\mathcal{L}_\mathsf{other}$ loss, we define "other" to be
> any sample whose label is not a descendant of the current node. This is
> utilized to calibrate our model and improve OOD detection performance.
>
> [1] Nalisnick E., Matsukawa A., Teh YW, Gorur D., Lakshminarayanan B. "Do deep generative models know what they don't know?" ICLR, 2018.
>
> [2] Morningstar W., Ham C., Gallagher A., Lakshminarayanan B., Alemi A., Dillon J. "Density of states estimation for out-of-distribution detection." AISTATS, 2021.

---

> > ### Author Response · Authors · 2022-11-14
> > **Response cont.**
> >
> > >The paper seems both to be concerned with the central ID / OOD question (which lacks some clarity) and the loosely related question of the performance of flat versus hierarchical classifiers.
> >
> > Finally, as you mention we consider ID/OOD as well as hierarchy vs. flat
> > classifiers. However, we argue that hierarchical classifiers can improve the
> > predictions provided to the end user by predicting at coarser granularities
> > when the model is uncertain which is not possible in flat classifiers. Again,
> > you identified that when the ID and OOD classes become more similar, it is
> > harder to determine what is (or should be) considered OOD. This question is
> > application dependent as each application will have different needs. For
> > example, a dog classifier may be required for self-driving car applications as
> > well as an ecological study. For the self-driving car, it is not necessary to
> > know the exact species rather it needs to know that its a dog and needs to be
> > avoided. On the other hand, in an ecological study looking to identify novel
> > dog breeds it is necessary to know if the species has been seen before. We utilize a
> > hierarchy to provide a framework for customizing the sensitivity of the
> > classifier to provide the user with the most useful information possible.
> > Therefore, hierarchical classification is closely related with fine-grain OOD detection.

---

### Official Review · Reviewer_Wf9S · 2022-11-01

**Confidence:** 4
**Correctness:** 3
**Technical Novelty And Significance:** 2
**Empirical Novelty And Significance:** 3
**Recommendation:** 5

**Clarity, Quality, Novelty And Reproducibility:**

Though the proposed evaluation setting and analysis is useful, because of lack of calibration & additional baselines, evaluations it is unclear interpret conclusive results under current evaluation. Also, it might be worthwhile to include a table for hierarchical methods to easily interpret overall results. Would encourage authors to include accuracy at each level and also percentage of in-correct, under-prediction, over-prediction & hierarchy distance for proposed method, MOS in a table.

**Strength And Weaknesses:**

Strengths:

Paper is well written, and presents good analysis by adopting & proposing various methods and metrics within in hierarchical classification such as over-prediction vs under-prediction & average hierarchy distance, etc. This is a useful direction in general to be considered for real-world use-cases. It is interesting to observe that there is no apparent benefit to visually derived hierarchies vs human-defined semantic hierarchy in current evaluation setting.

Weakness:

As the current method is fit based on likelihood in an autoregressive manner defined by hierarchy, it might important that at each level predictions are calibrated. Authors currently completely ignore calibration at intermediate levels which would result in more in-correct/over-predictions. It might be worthwhile for authors to consider calibration at each intermediate level. [3] also considers calibrated models at intermediate levels and also is evaluated for novelty detection within hierarchical classification framework, and it might be worthwhile to include this another baseline to consider.

Hierarchical classification is already adopted, and initial results are presented OOD setting in MOS baseline which authors included as one of baselines, and though proposed method is better in case of ImageNet 100 at fine-level its worse or on-par at coarse level especially for ImageNet1K.

Also authors it might be worthwhile to evaluate current method standard ImageNet openset detection benchmarks and also fine-grained/semantic OOD detection [1,2]

References: [1] https://eval.ai/web/challenges/challenge-page/1041/overview or https://www.cs.cmu.edu/~shuk/open-world-vision.html [2] Vaze et al. OPEN-SET RECOGNITION: A GOOD CLOSED-SET CLASSIFIER IS ALL YOU NEED? (ICLR 2022) Proposes imagenet benchmarks & other fine-grained OOD detection tasks. [3] K. Lee et al. Hierarchical Novelty Detection for Visual Object Recognition (CVPR 2018)

**Summary Of The Paper:**

In this work authors adopt hierarchical classification for OOD detection and based on hierarchy they make predictions at varying level of granularity which could provide enhance examinability w.r.t OOD. This is indeed useful for OOD detection or open-set detection, and authors provide various qualitative & quantitative analysis based on adopted hierarchical classification.

**Summary Of The Review:**

Authors present a good evaluation study by adopting hierarchical classification for OOD, but currently lacks novelty & further analysis would be required to interpret results/findings. I would encourage authors to address raised issues!

---

> ### Author Response · Authors · 2022-11-14
> **Response**
>
> Thank you for your thoughtful comments. Please see below for our response to each of your concerns.
>
> > As the current method is fit based on likelihood in an autoregressive manner defined by hierarchy, it might important that at each level predictions are calibrated. Authors currently completely ignore calibration at intermediate levels which would result in more in-correct/over-predictions.
>
> We agree that model calibration is important and necessary for good performance with hierarchical classifiers.
> We do not "completely ignore" this issue, but we utilize different terminology.
> Specifically, we note that the purpose of the "other" loss
> ($\mathcal{L}_\mathsf{other}$) discussed in Section 3.2 is to calibrate each of
> the internal node's classifiers by forcing it to predict a uniform distribution
> on images from the training dataset that are outside of its scope. The only
> classifier that is unaffected by this loss is the root node since all of the
> training set images are children of the root node. Note that supplemental
> outlier datasets could further aid in calibration by providing outliers at each
> level of the hierarchy including the root node. Furthermore, we mitigate the
> problem of incorrect predictions at early layers of the hierarchy (near the root)
> by up weighting the loss contribution of the early nodes as described in
> Section 3.2. Also, as noted in Wan et al. 2021, by adopting "soft" decisions we
> allow extremely confident leaf nodes to recover from an incorrect decision earlier in the hierarchy.
>
> >Hierarchical classification is already adopted, and initial results are presented OOD setting in MOS baseline which authors included as one of baselines, and though proposed method is better in case of ImageNet 100 at fine-level its worse or on-par at coarse level especially for ImageNet1K.
>
> Regarding the MOS performance comparison, our technique (HSC) performs on-par
> with MOS across all datasets. The main advantage of our technique over MOS is
> that HSC provides interpretable decision making and intermediate inferencing
> whereas MOS does not. MOS does not provide an interpretability benefit because
> it does not utilize an informative hierarchy. MOS splits the 1000 Imagenet classes
> into 8 groups that are necessarily large, covering many loosely related classes.
> This provides much less information and ability to perform intermediate inference than the WordNet
> hierarchies used in HSC. Also, MOS only functions on 2-level hierarchies which limits its applicability.
>
> > Also authors it might be worthwhile to evaluate current method standard ImageNet openset detection benchmarks and also fine-grained/semantic OOD detection
>
> We agree that it may be informative to benchmark hierarchical classifiers across
> additional OOD and open-set datasets. However, the suggested benchmarks are
> large scale and due to limited time and computational resources we are
> not able to complete the experiments.
> Also, we designed our ID-/OOD- datasets as some of the first large-scale OOD
> tasks which provide OOD samples across fine, medium, coarse, and far granularities.
> This is in contrast with the majority of OOD studies that utilize small scale datasets with OOD samples that are coarse or far from the ID set.
> For example, [1,2,3] utilize CIFAR-10/CIFAR-100 (small scale) and far-OOD
> datasets, MOS (Huang et al 2021) scales to Imagenet-1K, but only evaluates
> far-OOD performance.
> OpenOOD [4] utilizes MNIST, CIFAR-10/-100, as well as, ImageNet-1K with far-OOD and near-OOD, however, they do not split the near-OOD by granularity (fine, medium, coarse) and they utilize outside datasets (iNaturalist, Species, OpenImages) instead of creating Imagenet ID/OOD splits to comprise near-OOD test sets that may have non-semantic differences due to image collection techniques.
> We plan to run additional experiments such as the ones suggested and open-source our code and results.
>
> > Would encourage authors to include accuracy at each level and also percentage of in-correct, under-prediction, over-prediction & hierarchy distance for proposed method, MOS in a table.
>
> Finally, we include the requested information in the added tables in the Appendix (section A.13 and figures 14, 15, & 16).
>
> [1] Wei H., Xie R., Cheng H., Feng L., An B., Li Y. "Mitigating neural network overconfidence with logit normalization." ICML, 2022.
>
> [2] Cao S., Zhang Z. "Deep hybrid models for out-of-distribution detection." CVPR, 2022.
>
> [3] Sun Y., Ming Y., Zhu X., Li Y. "Out-of-distribution detection with deep nearest neighbors." ICML, 2022.
>
> [4] Yang J., Wang P., Zou D., Zhou Z., Ding K., Peng W., Wang H., Chen G., Bo L., Sun Y., Du X., Zhou K., Zhang W., Hendrycks D., Li Y., Liu Z. "OpenOOD: Benchmarking generalized out-of-distribution detection." NeurIPS, 2022.

---

> ### Comment · Reviewer_Wf9S · 2022-12-12
> **Response to Authors**
>
> I agree with authors that proposed method give interpretability without loss of performance but based on results for many setting the optimal hierarchy still seems to be very small like 2.  Yes, my comment was more about explicit calibration.
>
> But similar to previous comment, as it currently stands  besides some improved interpretability which is partially limited by depth of hierarchy without loosing OoD detection performance the overall novelty, contribution & insights from the paper seem rather limited & I still think its borderline.

---

### Decision · Program_Chairs · 2023-01-20

**Decision:**

Reject

**Justification For Why Not Higher Score:**

Limited Novelty as several papers have already investigated hierarchical outlier detection
Contributions are only marginally significant

**Justification For Why Not Lower Score:**

N/A

**Metareview: Summary, Strengths And Weaknesses:**

The paper proposes fine-grain inference for hierarchical OOD detection, i.e. at varying levels of granularity.
The reviewers found the paper well-written. The main concerns were the following:
- "Hierarchical novelty detection" and "hierarchical outlier detection" have been already investigated in several previous papers, hence the novelty appears to be a bit limited.
- It's not clear if/how the proposed approach is superior to existing approaches (cf. the Imagenet-1k experimental results in Table 1).
- Note: Reviewer NPqs flagged concern about OOD terminology, but that's not really specific to this paper, so I ignored that in my decision.

Overall, the reviewers felt that the contributions are only marginally significant, and the current version falls below the acceptance threshold. I encourage the authors to revise the draft based on reviewers' feedback and resubmit to another venue.